# TANGO1 builds a machine for collagen export by recruiting and spatially organizing COPII, tethers and membranes

Ishier Raote[1,2†], Maria Ortega-Bellido[1,2†], António JM Santos[1,2‡], Ombretta Foresti[1,2], Chong Zhang[3], Maria F Garcia-Parajo[4,5], Felix Campelo[4], Vivek Malhotra[1,2,5*]

[1]Centre for Genomic Regulation, The Barcelona Institute of Science and Technology, Barcelona, Spain; [2]Universitat Pompeu Fabra, Barcelona, Spain; [3]SIMBIOsys Group, Department of Information and Communication Technologies, Universitat Pompeu Fabra, Barcelona, Spain; [4]ICFO-Institut de Ciencies Fotoniques, The Barcelona Institute of Science and Technology, Castelldefels, Spain; [5]Institució Catalana de Recerca i Estudis Avançats, Barcelona, Spain

*For correspondence:
vivek.malhotra@crg.eu

†These authors contributed equally to this work

Present address: ‡Division of Hematology, Department of Medicine, Stanford University, Stanford, United States

**Abstract** Collagen export from the endoplasmic reticulum (ER) requires TANGO1, COPII coats, and retrograde fusion of ERGIC membranes. How do these components come together to produce a transport carrier commensurate with the bulky cargo collagen? TANGO1 is known to form a ring that corrals COPII coats, and we show here how this ring or fence is assembled. Our data reveal that a TANGO1 ring is organized by its radial interaction with COPII, and lateral interactions with cTAGE5, TANGO1-short or itself. Of particular interest is the finding that TANGO1 recruits ERGIC membranes for collagen export via the NRZ (NBAS/RINT1/ZW10) tether complex. Therefore, TANGO1 couples retrograde membrane flow to anterograde cargo transport. Without the NRZ complex, the TANGO1 ring does not assemble, suggesting its role in nucleating or stabilising this process. Thus, coordinated capture of COPII coats, cTAGE5, TANGO1-short, and tethers by TANGO1 assembles a collagen export machine at the ER.

DOI: https://doi.org/10.7554/eLife.32723.001

## Introduction

As secretory cargoes increase in size and complexity through evolution, mechanisms for their export from the endoplasmic reticulum (ER) must adapt concomitantly. Collagens, the most abundant secretory cargo in mammals - representing nearly 25% of the dry weight of the mammalian body, are some of the most challenging of all secretory cargoes (*Kadler et al., 2007*). Several requirements make collagen secretion a challenging task. *First*, in a complex multi-step process, collagens in the ER fold and trimerise into rigid, rod-like elements (*Ishikawa et al., 2015*; *Kadler, 2017*) of up to 400 nm in length (*Burgeson et al., 1985*). The folding/assembly of collagen must be coupled to its export, to retain unassembled collagen in the ER, whilst ensuring that all rod-like fully assembled collagen is rapidly exported. *Second*, assembled collagens are too large to fit into generic COPII-coated vesicles that are usually less than 90 nm in diameter (*Malhotra and Erlmann, 2015*; *Miller and Schekman, 2013*). *Third*, the rapidity with which this cargo exits the ER and passes through the secretory pathway, requires efficient transfer between compartments.

Our identification (*Bard et al., 2006*; *Saito et al., 2009*) and the subsequent characterisation of TANGO1 (*8–11*) has revealed a single protein, conserved through most metazoans, that stands at the crossroads of all these processes, modulating them to bring about bulky cargo export from the ER. TANGO1 is an ER exit site (ERES)-localized, transmembrane protein required for export of

collagen and other bulky protein components of the extracellular matrix such as Dumpy (*Saito et al., 2009*; *Ríos-Barrera et al., 2017*; *Maiers et al., 2017*; *Tomoishi et al., 2017*). *Figure 1* is a schematic of three TANGO1 family proteins: TANGO1, TANGO1-short and cTAGE5. A brief description of these proteins follows.

TANGO1 is a protein of 1907 amino acids (*Figure 1A*) of which 709 face the cytoplasm. TANGO1 contains a full transmembrane domain and a second membrane-inserted loop, which partially inserts into the inner leaflet of the ER membrane. The lumenal part contains a coiled-coil domain and, at the N terminus, an SH3-like domain. The SH3-like domain binds collagens via HSP47 (*Saito et al., 2009*; *Ma and Goldberg, 2016*; *Maeda et al., 2017*; *Wilson et al., 2011*). The cytoplasmic part of TANGO1 is composed of two coiled-coil domains (CC1 and CC2) followed by a C-terminal proline-rich domain (PRD). CC1 contains a domain called TEER (Tether for ERGIC at the ER) that recruits ERGIC-53-containing membranes (*Santos et al., 2015*); CC2 binds cTAGE5 (*18*), and PRD binds Sec23 and Sec16 (*Saito et al., 2009*; *Ma and Goldberg, 2016*).

TANGO1-short is a spliced isoform of TANGO1. It is composed of 785 amino acids that arise from the same exons that encode the cytoplasmic domains of TANGO1. The sequence of TANGO1-short differs in the membrane-inserted helix, and it contains only 15 amino acids at the N terminus, within the ER lumen. It therefore lacks any capacity to interact directly with cargoes. We expect that TANGO1-short binds the same cytoplasmic proteins as TANGO1, but this has not been directly tested.

Evolutionarily, TANGO1 appears to have been duplicated early in metazoans, yielding a TANGO1-like protein (TALI) (*Bosserhoff et al., 2003*). Like TANGO1, TALI is expressed as two isoforms. The long isoform is expressed in select tissues while the short isoform (cTAGE5) has a ubiquitous expression (*Santos et al., 2016*; *Bosserhoff et al., 2003*; *Pitman et al., 2011*; *Pfeffer, 2016*). cTAGE5 is composed of 804 amino acids, with a short lumenal stretch of 38 amino acids, followed by a single transmembrane domain. The organisation of cytoplasmic domains is the same as TANGO1, with two coiled-coil domains and a PRD. The first (CC1) of cTAGE5 interacts with Sec12; CC2 interacts with TANGO1 and Sec22, and the PRD, like TANGO1, interacts with Sec23 (*Wilson et al., 2011*; *Saito et al., 2011*; *Saito et al., 2014*; *Raote et al., 2017*).

From the published data on these proteins, we can conclude that all three family members bind each other and Sec23. cTAGE5 binds Sec12 and Sec22. TANGO1 (and therefore TANGO1-short) does not bind Sec12. Of these proteins, only TANGO1 can bind cargo in the lumen. How different binding partners could affect the overall function of these proteins in ERES assembly and cargo export, remains untested.

A newly discovered feature of TANGO1 is its lateral organisation into rings of up to 300 nm diameter, which corral COPII coats at the ERES (*Raote et al., 2017*). The organisation of cTAGE5 and TANGO1-short in TANGO1 rings is not known.

Exploiting the modular composition of TANGO1, we have generated forms of TANGO1 (*Figure 1—figure supplement 1A*), each missing one specific domain and hence with one specific set of functions/interactions abrogated. With this set of reagents, we now address how TANGO1 assembles into a functional ring or a fence. We show that this fence of TANGO1 family proteins surrounds COPII, and through specific tethers, physically links the ER and ERGIC for collagen export.

## Results

### Binding of TANGO1 to COPII controls TANGO1 ring formation

The role of COPII in TANGO1 ring assembly could be addressed by using a mutant form of TANGO1 that lacks the PRD (TANGO1ΔPRD), which therefore cannot interact with Sec23 (*Saito et al., 2009*) (schematic of TANGO1, *Figure 1A*). 2H5 cells (HeLa cells with TANGO1 deleted using the CRISPR/Cas9 system [*Santos et al., 2015*]) were co-transfected with collagen VII and either TANGO1 or TANGO1ΔPRD and imaged using STED microscopy. Full length TANGO1 formed distinct rings of somewhat uniform shape and size (*Figure 2A*). Surprisingly, TANGO1ΔPRD also assembled into rings, but with two clear differences. First, rings were smaller (*Figure 2B*, *Figure 2—figure supplement 1A*); and second, some rings appeared fused with each other to form either a planar tessellation (*Figure 2C*, *Figure 2—figure supplement 1G*) or long linear assemblies (*Figure 2D*, *Figure 2—figure supplement 1B–F*). Quantitative morphological descriptors of the size and shape of

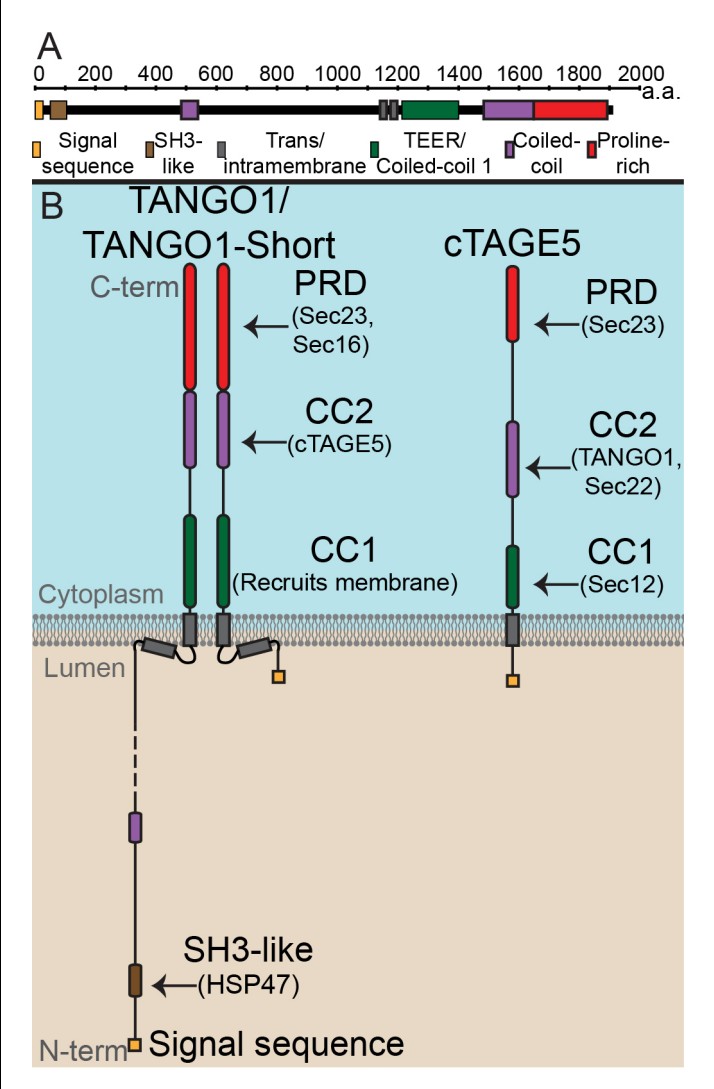

**Figure 1.** The domain architecture and topology of TANGO1 and cTAGE5. (**A**) A schematic depiction of full length TANGO1, showing the extent of each domain in amino acids. (**B**) Three TANGO1-family proteins (TANGO1, TANGO1-short and cTAGE5) that form a stable complex at the ERES (*Maeda et al., 2016*). TANGO1 is a type one single-pass transmembrane protein of 1907 amino acids, localised to ER exit sites. TANGO1 has an N-terminal lumenal SH3-like domain that interacts with collagen (*Saito et al., 2009*) via the chaperone, HSP47 (*Ishikawa et al., 2016*). There is a transmembrane helix and, in close proximity, a membrane insertion helix. On the cytoplasmic side of the ER membrane, TANGO1 has two coiled-coil (CC) domains (CC1 and CC2). CC1 is used by TANGO1 to recruit ERGIC membranes for producing a collagen carrier (*Santos et al., 2015*). CC2 binds to a similar coiled-coil domain in cTAGE5 (*18*). The proline-rich domain (PRD) binds ER exit site machinery Sec23 (*Saito et al., 2009*; *Ma and Goldberg, 2016*) and Sec16 (*Maeda et al., 2017*). Alternative splicing of TANGO1 results in a short isoform, TANGO1-short (*Wilson et al., 2011*), lacking the lumenal domain. The closely related protein cTAGE5 has a similar cytoplasmic domain organisation with two coiled-coil domains (CC1 and CC2) and a proline-rich domain (PRD). Via its CC1 it recruits Sec12 (*Saito et al., 2014*). cTAGE5 and TANGO1/TANGO1-short interact through their respective CC2 domains. In addition, the cTAGE5 CC2 also interacts with the retrograde v-SNARE Sec22 (*Fan et al., 2017*). Like the TANGO1/TANGO1-short PRDs, the cTAGE5 PRD also interacts with Sec23 (*Ma and Goldberg, 2016*; *Saito et al., 2011*; *Wang et al., 2016*).

DOI: https://doi.org/10.7554/eLife.32723.002

The following figure supplement is available for figure 1:

**Figure supplement 1.** Constructs used in this study.
DOI: https://doi.org/10.7554/eLife.32723.003

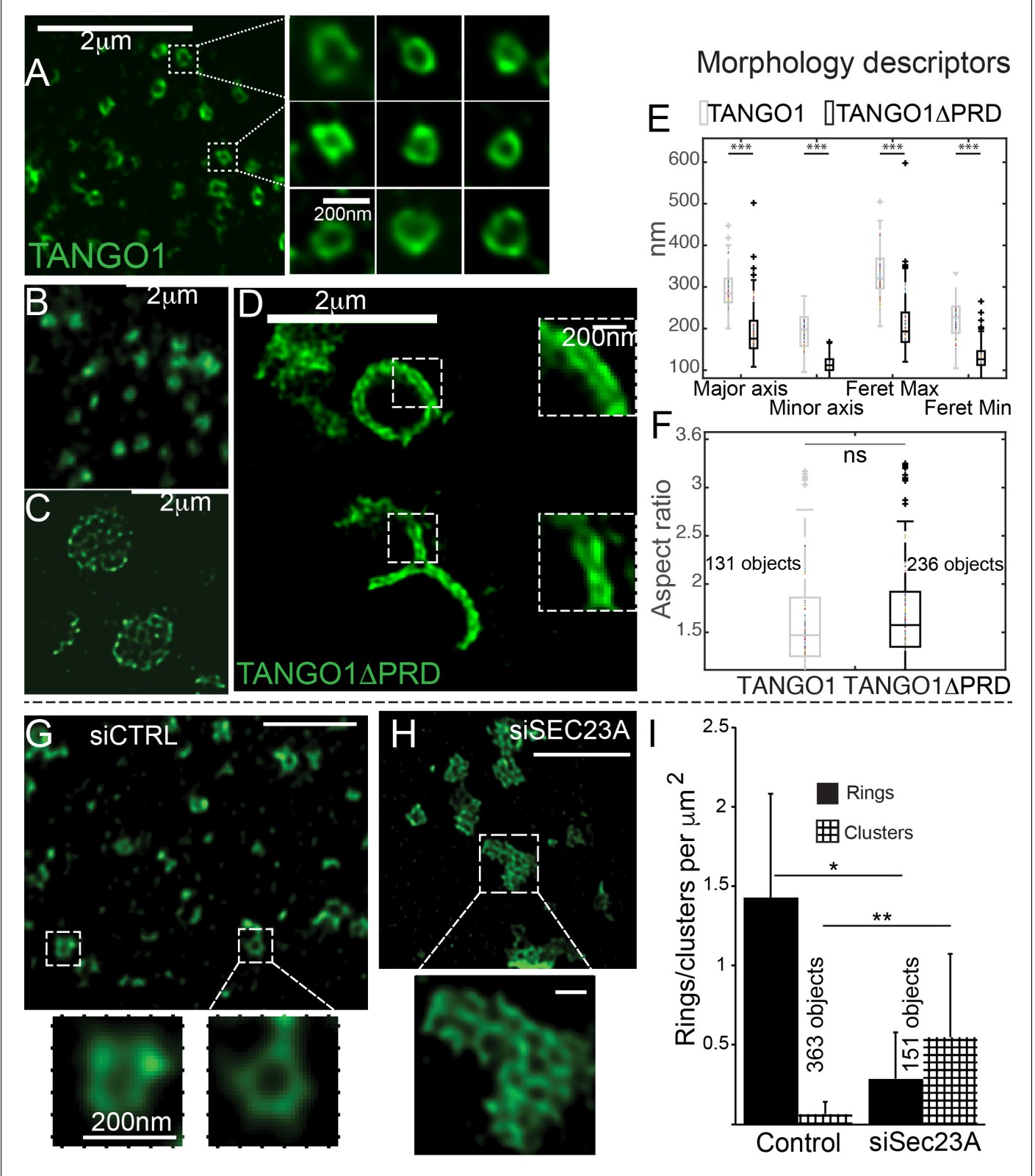

**Figure 2.** The role of COPII in assembly of TANGO1 into rings. In TANGO1 knockout cells, various constructs of TANGO1 were expressed and visualized by STED microscopy. (**A**) Full-length TANGO1 forms rings. (**B**) TANGO1ΔPRD forms small distinct rings, (**C**) rings fused into a planar tessellation or (**D**) rings fused in long linear rows. (**E**) Quantification of size, shown as a scatter plot and box plot of measured morphological descriptors: major axis and minor axis, diameter of a fitted ellipse, maximum and minimum Feret's diameters. Quantification of shape, shown as a

*Figure 2 continued on next page*

*Figure 2 continued*

scatter and box plot (**F**) of the aspect ratio between the major and minor axes. 131 and 236 rings respectively were analysed for the two constructs. STED images of TANGO1 in siCTRL (**G**) or siSEC23A (**H**) treated RDEB/FB/C7. (**I**) Rings (solid bar) or clusters (checkered bar) in 22 siCTRL cells and 14 siSEC23A cells were manually counted and plotted, normalised to the area of collagen accumulations. (**E**) ***p<0.006; ns not significant. (**I**) **p<0.01, *p<0.05 (Student's t test). Scale bars (**A–D, G, H**) 2 µm, insets 200 nm.

DOI: https://doi.org/10.7554/eLife.32723.004

The following figure supplements are available for figure 2:

**Figure supplement 1.** Structures formed by TANGO1ΔPRD.

DOI: https://doi.org/10.7554/eLife.32723.005

**Figure supplement 2.** Image analysis workflow.

DOI: https://doi.org/10.7554/eLife.32723.006

**Figure supplement 3.** SEC23A is required for collagen secretion.

DOI: https://doi.org/10.7554/eLife.32723.007

**Figure supplement 4.** Structures formed by TANGO1 after depletion of SEC23A.

DOI: https://doi.org/10.7554/eLife.32723.008

structures formed by TANGO1 constructs, were extracted using semi-automated image analysis (*Figure 2—figure supplement 2*, *Table 1*) and are described in detail in the Materials and methods section and the figure legend. Specifically, we fitted rings to an elliptical shape and measured the diameters of the ring in terms of major and minor axes of its fitted ellipse. This works well for regular-shaped ellipses, however for structures and shapes that deviate from an elliptical shape, a rectangular bounding shape is a more useful approximation. Therefore, maximum and minimum diameters (Feret's maximum or minimum) were also extracted and all these values are plotted in *Figure 2E*. From this quantification, we confirmed that rings formed by TANGO1ΔPRD, are significantly smaller than rings formed by TANGO1 (*Figure 2E*, *Table 1*). We used the aspect ratio (the ratio of the major to minor axes of the fitted ellipses) as a descriptor of the shape of rings. By this measure, rings formed by TANGO1 and TANGO1ΔPRD had a similar shape (*Figure 2F*).

It is important to note that these cells still contain TANGO1-short and cTAGE5 (*Figure 1*), both of which will recruit TANGO1ΔPRD to ERES. These data suggest that the cytoplasmic domains of the TANGO1-family of proteins act as a single unit and any one can assemble into a ring, however TANGO1 brings cargo to the exit site. This suggests that overexpressing cytoplasmic isoforms (either TANGO1-short or cTAGE5) would increase the capacity of an ERES to export cargo, however TANGO1 is the only protein with the capacity to bring cargo to ERES. Collagen secreted in the absence of TANGO1 might thus be in an unfolded or unassembled form.

In a complementary experiment, we studied the effect of Sec23A depletion on TANGO1 ring formation in RDEB/FB/C7 fibroblasts. Depleting cells of all Sec23 could create cellular stress and affect endomembrane regulation, so we attempted to minimise such a potential stress by using siRNA that targeted exclusively Sec23A, and not Sec23B. As expected, collagen export from the ER was reduced in Sec23A-depleted cells (*Figure 2—figure supplement 3*).

While TANGO1 in control cells was often visualised in rings (*Figure 2G*), depletion of Sec23A appeared to phenocopy our results with TANGO1ΔPRD, showing multiple seemingly fused rings of TANGO1 assembled in planar arrays (*Figure 2H*, *Figure 2—figure supplement 4*), quantified in *Figure 2I*. These structures/abnormal rings were almost never observed in cells expressing full length TANGO1, or cells that are not depleted of Sec23A.

**Table 1.** Quantification of the size and shape of rings formed by TANGO1 and its mutant forms.

| | Major axis (nm) | Minor axis (nm) | Feret's major axis (nm) | Feret's minor axis (nm) | Objects counted | Cells imaged |
|---|---|---|---|---|---|---|
| TANGO1 | 293 ± 47 | 191 ± 43 | 330 ± 53 | 221 ± 46 | 131 | 44 |
| TANGO1ΔPRD | 192 ± 55 | 115 ± 19 | 210 ± 59 | 130 ± 29 | 236 | 40 |
| TANGO1ΔCC2 | 313 ± 77 | 164 ± 50 | 358 ± 90 | 203 ± 56 | 228 | 51 |

DOI: https://doi.org/10.7554/eLife.32723.009

Based on our super-resolution microscopy images, we hypothesise that TANGO1 rings could be represented as a multimeric assembly of units of TANGO1 family proteins (TANGO1, TANGO1-short and cTAGE5) that assemble into a fence.

## Lateral interactions along the circumference of a TANGO1 ring

A key feature that could provide strength to a fence of TANGO1 would be lateral interactions between components in the fence. For example, the TANGO1-interacting protein cTAGE5 (*Figure 3A*) should be a component of the ring and could contribute to lateral interactions in the ring. We visualised TANGO1 and cTAGE5 in RDEB/FB/C7 cells by STED microscopy. Due to the low quality of commercially available anti-cTAGE5 antibodies for immunofluorescence, we were unable to visualise the localisation of cTAGE5 as clearly as TANGO1, nonetheless cTAGE5 clearly localised along the rings delineated by TANGO1 (*Figure 3B*).

To test the involvement of cTAGE5 in TANGO1 ring formation, we generated a construct of TANGO1 lacking the second cytoplasmic coiled-coil (TANGO1ΔCC2) domain (*Figure 1—figure supplement 1* for a schematic) and hence, unable to interact with cTAGE5 (*Figure 3A*). STED microscopy revealed that, in contrast to full length TANGO1 (*Figure 3C*), TANGO1ΔCC2 assembled into misshapen structures (*Figure 3D* and *Figure 3—figure supplement 1*). Ring size and shape were quantified as in the previous section. Rings formed by TANGO1ΔCC2 were more variable in size (*Figure 3E*, *Table 1*) and shape (*Figure 3F*) than those formed by full length TANGO1.

As a complementary approach, we characterised the effect of depleting cTAGE5, on ring formation in cells with endogenous TANGO1. As expected, in RDEB/FB/C7 fibroblasts depleted of cTAGE5 (*Figure 3—figure supplement 2A*), collagen secretion was blocked (*Figure 3—figure supplement 2B,C*). TANGO1 structures phenocopied TANGO1ΔCC2 structures in 2H5 cells: rings of TANGO1 were misassembled (*Figure 3G*) and formed unusual shapes, without considerably altering the number of rings observed (*Figure 3H*).

Another lateral interaction that might maintain fence integrity could be an intrinsic ability of TANGO1 to self-associate. A test of this proposition would be to identify a domain in TANGO1 that mediates self-association and show that it has a role in ring formation. To identify such a domain, we tested the ability of TANGO1-FLAG to co-immunoprecipitate with TANGO1ΔPRD, TANGO1ΔCC2 or TANGO1ΔCC1 (*Figure 1—figure supplement 1*). We observed (*Figure 4A*) that TANGO1-FLAG was immunoprecipitated by TANGO1 and TANGO1ΔPRD, but not by TANGO1ΔCC2 (*Figure 4A*) or TANGO1ΔCC1 (*Figure 4B*). Reasoning that the effect of the CC2 was likely indirect, as TANGO1ΔCC2 is unable to interact with cTAGE5 (*Figure 4A*) (*Saito et al., 2011*; *Saito et al., 2014*), we focused on the first coiled-coil domain (CC1) to identify a minimal region required for self-association. We generated two TANGO1 constructs with smaller deletions from the CC1, each of which had a deletion in a portion of the coiled-coil (TANGO1Δ1255–1295 and TANGO1Δ1296–1336). As a control, we confirmed these constructs still interacted with cTAGE5 (*Figure 4B*). Only TANGO1Δ1255–1295 did not immunoprecipitate TANGO1-FLAG (*Figure 4B*).

With a minimal self-association domain (a.a. 1255–1295) identified, we looked for its role in TANGO1 ring formation. 2H5 cells were co-transfected with collagen VII and either TANGO1ΔCC1, TANGO1Δ1255–1295 or TANGO1Δ1296–1336 and then imaged by STED microscopy. In line with our predictions, TANGO1ΔCC1 or TANGO1Δ1255–1295 could not form rings; of the 16 and 15 cells examined respectively, there were few discernible polymeric assemblies of TANGO1 (*Figure 4C,D*), while TANGO1Δ1296–1336 behaved as full length TANGO1, forming distinct, readily detectable, independent rings (*Figure 4E*) of similar size (*Figure 4—figure supplement 1A*) and shape (*Figure 4—figure supplement 1B*) as TANGO1. These data indicate that TANGO1-TANGO1 interactions (*Figure 4F*), mediated by amino acids 1255–1295, are required to maintain ring integrity.

In our coarse-grained view of this fence of TANGO1 and TANGO1 family of proteins (cTAGE5 and TANGO1-short), we would describe our data thus far in terms of two general sets of interactions. First, lateral interactions mediated by TANGO1 self-association and its interaction with cTAGE5 and TANGO1-short, and second, inward attractions of TANGO1/cTAGE5/TANGO1-short to COPII, thus affecting the ring size and its placement with respect to COPII budding machinery.

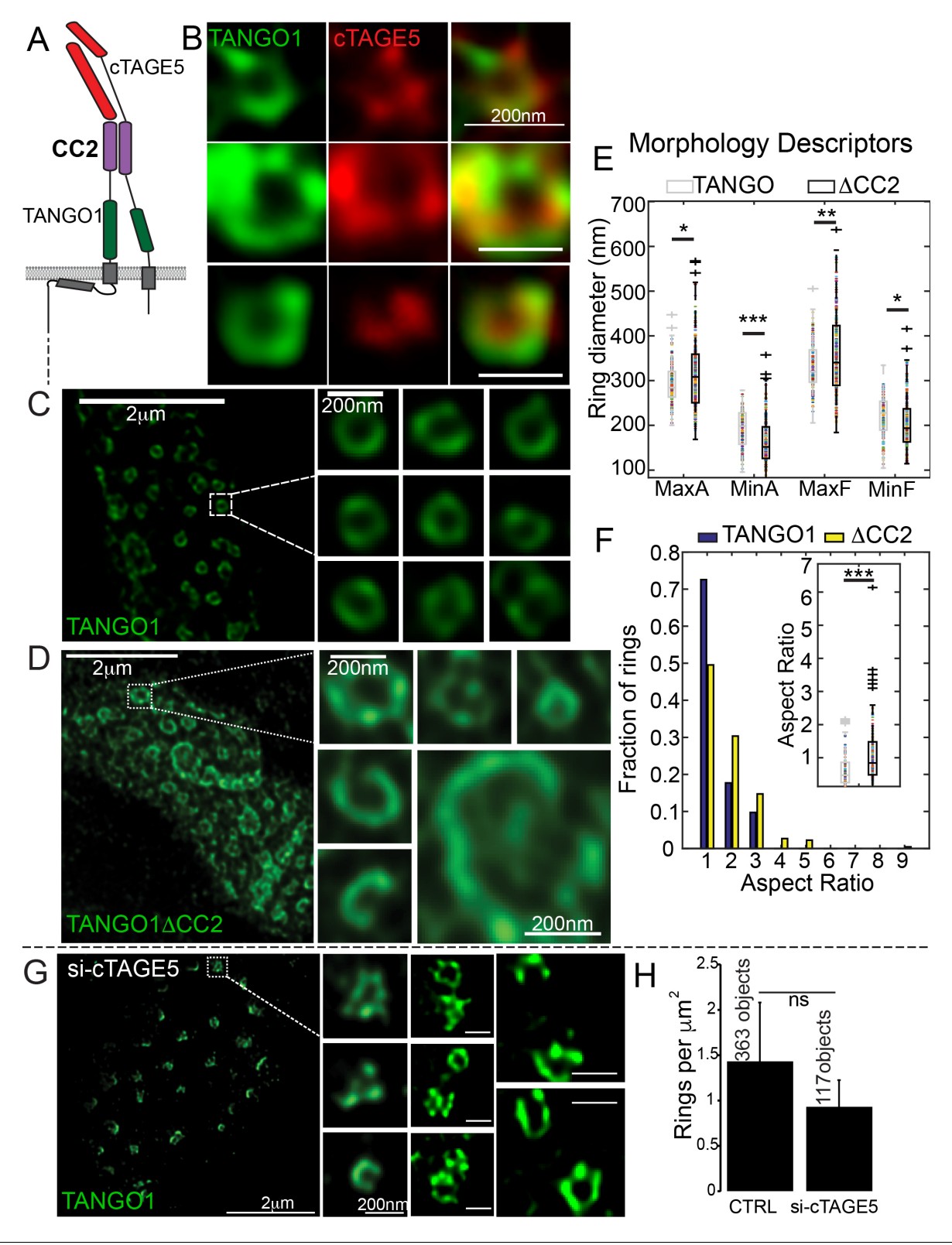

**Figure 3.** Lateral interactions in TANGO1 ring assembly mediated by cTAGE5. (**A**) Schematic of the interaction of TANGO1 and cTAGE5. (**B**) STED images of TANGO1 and cTAGE5 in RDEB/FB/C7. 70 rings were manually counted from 12 cells and scored for cTAGE5 signal localisation within the ring. 21 rings showed peripherally located cTAGE5 while 49 had cTAGE5 within the ring formed by TANGO1. Rings of TANGO1 (**C**) and TANGO1ΔCC2 (**D**) in 2H5 cells. (**E**) scatter and box plots of measured morphological size descriptors: major and minor axes diameters of fitted ellipse (MaxA, MinA),

*Figure 3 continued on next page*

*Figure 3 continued*

and Feret's diameter (MaxF, MinF). (**F**) Binning rings of TANGO1 (blue bars) and TANGO1ΔCC2 (yellow bars) by aspect ratio (major to minor axes of the fitted ellipse). Inset, quantification of shape, shown as scatter-plot and box plot of the aspect ratio. The number of rings analysed for the independent experiments are 131 and 228, respectively. (**G**) STED image of TANGO1 in si-cTAGE5 in RDEB/FB/C7. (**H**) Quantification of number of rings observed in control cells (22 cells) or si-cTAGE5 cells (13 cells) normalised to the area of collagen accumulations. Scale bars (**B**) 200 nm; (**C, D**) 2 μm, insets 200 nm (**G**) 1 μm; insets 200 nm, *p<0.05; **p<0.01; ***p<0.001, ns not significant.

DOI: https://doi.org/10.7554/eLife.32723.010

The following figure supplements are available for figure 3:

**Figure supplement 1.** Structures formed by TANGO1ΔCC2.

DOI: https://doi.org/10.7554/eLife.32723.011

**Figure supplement 2.** cTAGE5 is required for collagen secretion.

DOI: https://doi.org/10.7554/eLife.32723.012

## Compartment tethering in a TANGO1 ring assembly pathway

We have shown recently that TANGO1, via its CC1, recruits ERGIC membranes that fuse at the ERES (*Santos et al., 2015*). Could TANGO1 rings concentrate membrane recruitment for mega-carrier biogenesis? What role does the TEER domain play in ring assembly? To address these questions, we first identified a minimal TEER domain within the CC1, using our previously developed approach (*Santos et al., 2015*).

Following our previous methodology (*Santos et al., 2015*), we generated two myc-tagged, mitochondrially-targeted TEER (mit-TEER truncates) constructs of 82 and 81 amino acids, respectively. Our original construct (*Santos et al., 2015*) had TANGO1 amino acids 1188 to 1396. From this, we generated two smaller constructs. In one, we deleted amino acids 1255–1295 (mit-Δ1255–1295); while in the other we deleted amino acids 1296–1336 (mit-Δ1296–1336) (*Figure 5A*). These corresponded exactly to the deletions in the CC1 described in the previous section.

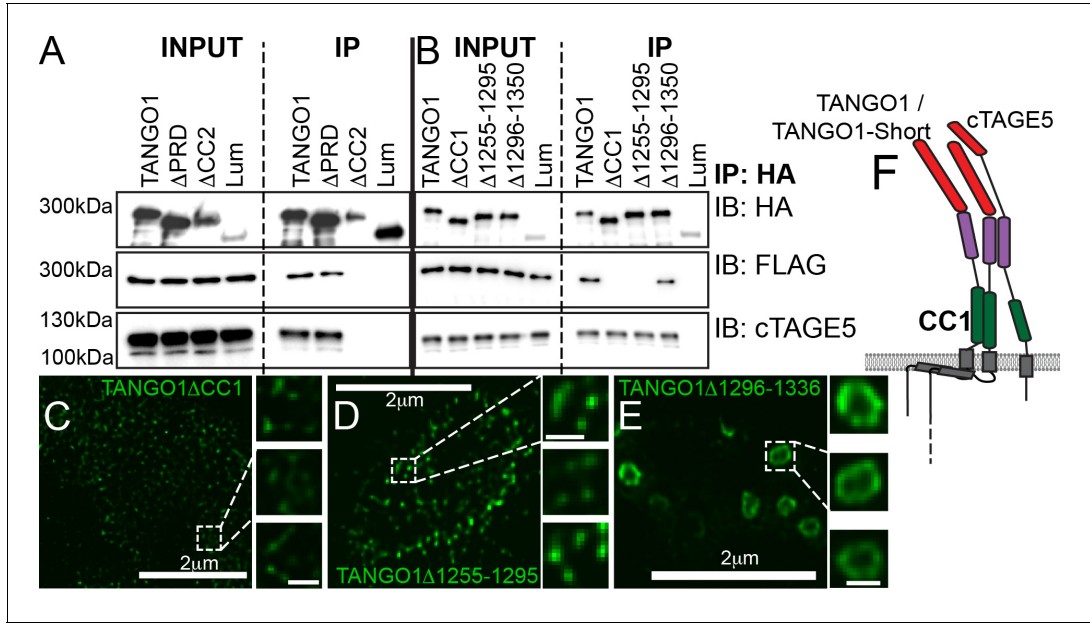

**Figure 4.** Lateral interactions in TANGO1 ring assembly mediated by TANGO1 self-association. (**A, B**) Co-immunoprecipitation of TANGO1-FLAG with the indicated constructs in HEK293T cells. Lysates and immunoprecipitated samples were probed for HA, FLAG and cTAGE5. 2H5 cells co-transfected with collagen VII and (**C**) TANGO1ΔCC1 (16 cells imaged), (**D**) TANGO1Δ1255–1295 (15 cells imaged) or (**E**) TANGO1Δ1296–1336 (16 cells imaged), were imaged by STED microscopy. (**F**) Schematic of interactions between TANGO1, TANGO1-short and cTAGE5. Scale bars (**C–E**) 2 μm; insets 200 nm.

DOI: https://doi.org/10.7554/eLife.32723.013

The following figure supplement is available for figure 4:

**Figure supplement 1.** Morphological quantification of structures formed by TANGO1Δ1296–1336.

DOI: https://doi.org/10.7554/eLife.32723.014

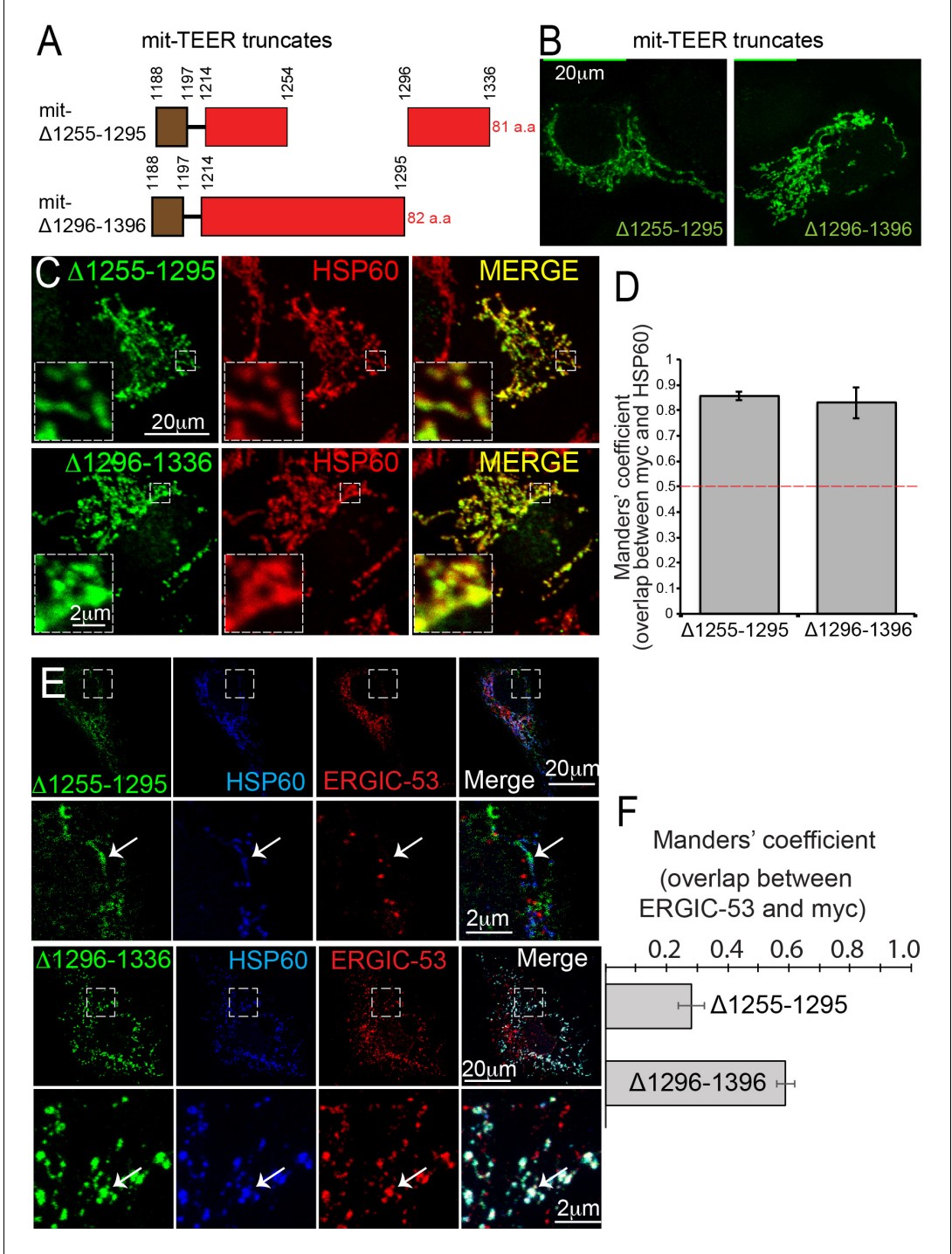

**Figure 5.** TANGO1 amino acids 1255–1295 are the minimal TEER. (A) A schematic depiction of myc-epitope tagged mitochondrially-targeted (mit-TEER) truncates. (B) mit-TEER truncates were expressed in 2H5 cells, fixed and stained with anti-myc-antibody and visualised with confocal microscopy. (C) mit-TEER truncates were expressed in 2H5 cells, which were fixed and stained using anti-myc antibody (green) and, as a mitochondrial marker, anti-HSP60 antibody (red). (D) Overlap of the signal from myc and HSP60 was quantified and plotted as the Manders' overlap coefficient for the two constructs (mit-Δ1255–1295 and mit-Δ1296–1336 respectively). (E) 2H5 cells were transfected with mit-Δ1255–1295 or mit-Δ1296–1336, fixed, and stained with anti-myc, anti-HSP60 and anti-ERGIC-53 antibodies. Arrows indicate myc staining with or without colocalised ERGIC-53 staining. (F) The extent of overlap of ERGIC-53 and myc was quantified and plotted as the Manders' overlap coefficient for mit-Δ1255–1295 and mit-Δ1296–1336, respectively. Scale bars: (B, C, E and F) 20 μm; inset 2 μm.

DOI: https://doi.org/10.7554/eLife.32723.015

We expressed the constructs in HeLa cells, fixed and then stained them using an anti-myc antibody and visualised these samples using confocal microscopy (*Figure 5B*). We confirmed the two constructs co-localised with the mitochondrial marker HSP60 (*Figure 5C*). The extent of overlap of myc-epitope and HSP60 was quantified and is plotted as the Manders' overlap coefficient (*Figure 5D*).

As before (*Santos et al., 2015*), we co-stained transfected cells with anti-ERGIC-53 and anti-myc antibodies. To our surprise, mitochondria expressing mit-Δ1255–1295 showed no recruitment of ERGIC-53-containing membranes (*Figure 5E*). In contrast, mit-Δ1296–1336 still functioned as the TEER domain and recruited ERGIC membranes. The extent of colocalisation of ERGIC-53 and myc for the two constructs was quantified and is plotted as Manders' overlap coefficient (*Figure 5F*). This tells us that the minimal TEER is exactly the same forty amino acids we identified in the previous section, as those required for the self-association of TANGO1. This implies that either a TANGO1 dimer can recruit a tether or the tether links two TANGO1 monomers. This hypothesis is tested and presented in Figure 7.

But how does this minimal TEER domain recruit ERGIC membranes? A prime candidate for this tethering activity is the evolutionarily conserved NRZ (NBAS, RINT1, ZW10) protein tether. NRZ tether is a multi-subunit tether complex (MTC) that assembles at the surface of the ER (*Ren et al., 2009*), is required for retrograde capture of membranes (*Aoki et al., 2009*; *Arasaki et al., 2006*; *Hirose et al., 2004*), partially localises to ER exit sites (*Schröter et al., 2016*) and interacts with SNAREs that we have shown previously are required for collagen export from the ER (*Nogueira et al., 2014*; *Santos et al., 2015*). One component of the MTC (RINT1) was also identified in our screen for genes required for protein secretion (*Bard et al., 2006*). Mutations in another component NBAS, are linked to dysregulated collagen secretion in atypical osteogenesis imperfecta (*DDD Study et al., 2017*).

As in previous sections, we imaged TANGO1 in RDEB/FB/C7 cells, with Sec31 and RINT1 by confocal microscopy (*Figure 6—figure supplement 1*) and, by STED microscopy observed the tether protein RINT1 localised to one or two puncta at rings of TANGO1, occasionally adjacent to ERGIC-53-containing membranes (*Figure 6A* and *Figure 6—figure supplement 2*).

We transfected full-length TANGO1, TANGO1Δ1255–1295, TANGO1Δ1296–1336 or TANGO1-Lum (lumenal) in HEK293T cells and attempted to co-immunoprecipitate tether proteins. We saw that full length TANGO1 and TANGO1Δ1296–1336 immunoprecipitated all three of the proteins that form the tether (NBAS, RINT1, ZW10) (*Figure 6B*). This interaction was completely abrogated when we used TANGO1Δ1255–1295 (lacking the minimal TEER domain). As controls, we confirmed all constructs still interacted with cTAGE5 and TANGO1-Lum did not immunoprecipitate either tether proteins or cTAGE5 (*Figure 6B*).

Depleting TANGO1, NBAS or RINT1 from RDEB/FB/C7 fibroblasts inhibited collagen VII secretion (*Figure 6C–E*) and arrested collagen in the ER (*Figure 6C*). Does TANGO1 recruit ERGIC to intracellular collagen accumulations (*Santos et al., 2015*) via the NRZ tether? In cells depleted of RINT1, NBAS or TANGO1 (*Figure 6F,H*), we quantified ERGIC recruitment to accumulations of collagen in the ER. In all cases, ERGIC membrane recruitment was significantly reduced (*Figure 6G*).

These data showed a novel function of TANGO1, to recruit ERGIC membranes via the retrograde NRZ MTC to the ERES for collagen export. But is this function built into ring assembly?

In RDEB/FB/C7 depleted of RINT1, TANGO1 rings were completely disrupted (siCTRL vs. siRINT1 *Figure 7A* vs. B). We individually depleted each of the other two proteins in the MTC (NBAS or ZW10) and checked for the ability of TANGO1 to assemble into rings in RDEB/FB/C7 fibroblasts. As seen after depleting cells of RINT1, rings were observed far less frequently (quantified in *Figure 7C*). In all cases, ERES, as marked by TANGO1 and SEC31 are still formed (*Figure 7—figure supplement 1*).

There are at least two mechanistic possibilities that could link tether binding, the TANGO1 self-association domain, and ring formation. Either (a) the tether is required to hold together TANGO1 and TANGO1-short in the fence; or (b) complexes form with TANGO1/TANGO1 short and this dimer then recruits the tether, which stitches together a higher order structure, forming a fence.

We tested these hypotheses by performing sequential co-immunoprecipitations to look for TANGO1, TANGO1-short and cTAGE5 in a stable complex. Using lentiviral infections, we generated HEK293T cells stably expressing cTAGE5-FLAG and TANGO1-HA. We depleted these cells of individual NRZ tether proteins and then performed sequential immunoprecipitation, pulling first on

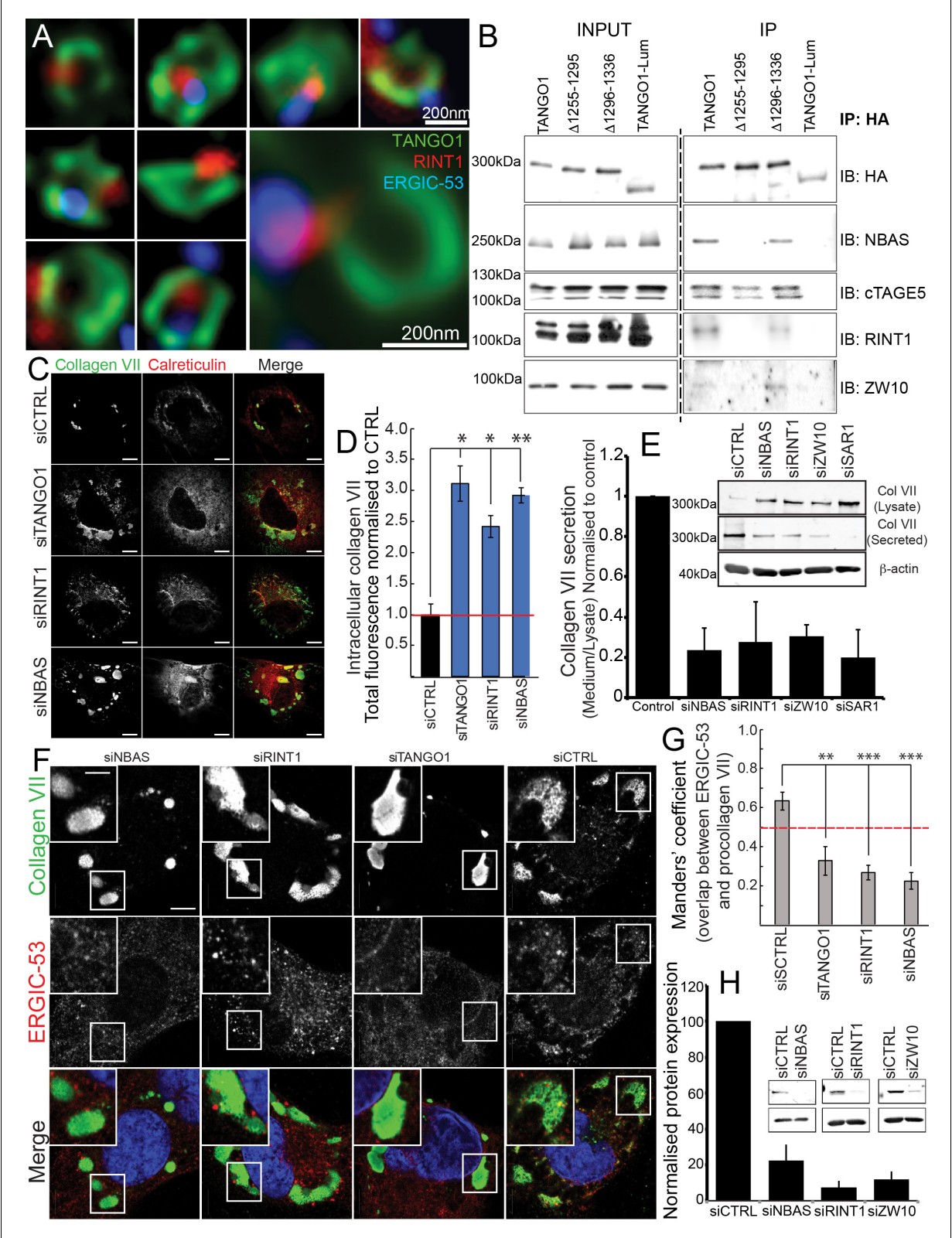

**Figure 6.** The NRZ tether links TANGO1 to ERGIC membranes. (**A**) Rings of TANGO1 (green) in RDEB/FB/C7 cells with RINT1 (red) and ERGIC-53 (blue). Deconvolved z-stacks of ten cells were used to quantify the location of the tether protein RINT1 relative to a ring of TANGO1. 90 rings of TANGO1 were manually scored, three adjacent slices in the image stack were used to identify signal from RINT1 in the vicinity of the ring of TANGO1. 23 rings showed RINT1 within the ring, 19 rings showed RINT1 on the circumference (at the edge) of the ring, 39 had RINT1 outside the ring, nine rings

*Figure 6 continued on next page*

*Figure 6 continued*

showed no detectable RINT1. (**B**) TANGO1, TANGO1Δ1255–1295, TANGO1Δ1296–1336 and TANGO1-Lum were expressed in HEK293T cells and immunoprecipitated. Samples were probed for NBAS, RINT1, cTAGE5 and ZW10. TANGO1 and TANGO1Δ1296–1336 immunoprecipitated all four proteins, but TANGO1Δ1255–1295 did not immunoprecipitate tether proteins. TANGO1-Lum did not pull down any of the four proteins. (**C**) RDEB/FB/C7 were transfected with siRNA (siCTRL, siNBAS, siRINT1 and siTANGO1) and immunostained for intracellular collagen VII (red) and calreticulin (green). (**D**) Quantification of fluorescence associated with intracellular collagen VII in (**C**). (**E**) Collagen VII secreted by RDEB/FB/C7 was looked at as the ratio of collagen in the medium to the lysate, quantified, and plotted as the average of values from at least three independent experiments. β-actin is a loading control. (**F**) siRINT1-, siNBAS- and siTANGO1-treated RDEB/FB/C7 were stained for collagen VII and ERGIC-53. (**G**) A plot of Manders' overlap coefficient for ERGIC-53 and collagen VII from (**F**) used to quantify ERGIC-53 localisation to collagen accumulations. (**H**) Representative blots showing the efficiency of knockdown of NBAS, RINT1 and ZW10, quantified and plotted as the average ±s.d. from at least three independent experiments. ***p<0.001; **p<0.01. Scale bars (**A**) 200 nm, (**C, F**) 10 μm, (**C**) inset) 5 μm.

DOI: https://doi.org/10.7554/eLife.32723.016

The following figure supplements are available for figure 6:

**Figure supplement 1.** RINT1 is recruited to exit sites at collagen accumulations.
DOI: https://doi.org/10.7554/eLife.32723.017

**Figure supplement 2.** RINT1 localises to one or two puncta in a TANGO1 ring.
DOI: https://doi.org/10.7554/eLife.32723.018

cTAGE5-FLAG and then TANGO1-HA and finally probed for TANGO1-short (*Figure 7D,F* for schematic). We observed that the NRZ tether had no effect on the association of TANGO1 and TANGO1-short in a stable complex (*Figure 7E*).

These data showed that the NRZ tether is required for TANGO1 to assemble into a ring and indicated that stable complexes of TANGO1, cTAGE5 and TANGO1-short, recruit the tether.

## Discussion

Our new data describe a mechanism whereby the very processes by which TANGO1 recruits ERES machinery and cargo, also bring about its own assembly into a fence of defined size. This in turn remodels the ERES, and in the lumen, via Hsp47, binds and potentially segregates assembled bulky cargoes (*Figure 8A*). Such a concerted mechanism circumvents a causality dilemma (the chicken-or-the-egg problem) in this process – neither ring nor function precedes the other; they assemble together, requiring each other to do so.

There are several broad implications of our data, addressing fundamental aspects of early secretory pathway organisation and cargo export.

### Tethering compartments

Tethers play a central role in membrane targeting and organelle biogenesis (*Cheung and Pfeffer, 2016*; *Gillingham and Munro, 2016*; *Munro, 2011*; *Pfeffer, 1999*; *Waters and Pfeffer, 1999*; *Wong and Munro, 2014*). Improved structural understanding has revealed fascinating models for the mechanisms of membrane recruitment by tethers (*Ren et al., 2009*; *Murray et al., 2016*). Our discovery of membrane recruitment by TANGO1 and its use of the NRZ tethering complex (*Figures 6* and *7*) has far reaching implications. A critical aspect of TANGO1 biology is that it functionally and physically couples anterograde to retrograde traffic at an ERES, coupling two successive compartments in the secretory pathway, allowing for more rapid and efficient cargo transport between the compartments (*Nogueira et al., 2014*; *Santos et al., 2015*; *Liu et al., 2017*). The NRZ tether would bind to, and recruit, any COPI-coated ERGIC-53-containing membranes in the vicinity of the ERES – but what of ERES closely apposed to the *cis*-Golgi, and what of organisms such as *D. melanogaster*, which have no discernible ERGIC compartment? Under such circumstances, the 'carrier' for collagen formed by the retrograde recruitment of COPI-coated membranes could just be the first Golgi cisterna. In other words, we could envisage a direct continuity or 'tunnel' between the ER and the Golgi (*Malhotra and Erlmann, 2015*), with a ring of TANGO1 and its associated exit site machinery holding together the two compartments, but also functionally delimiting them.

We have not observed a complete ring of tethers with TANGO1. The tethers instead appear as one or two puncta at the ring circumference. One can envisage that an initiation point of the TANGO1 ring recruits tethers and TANGO1 continues to assemble into a ring whereas the tethers

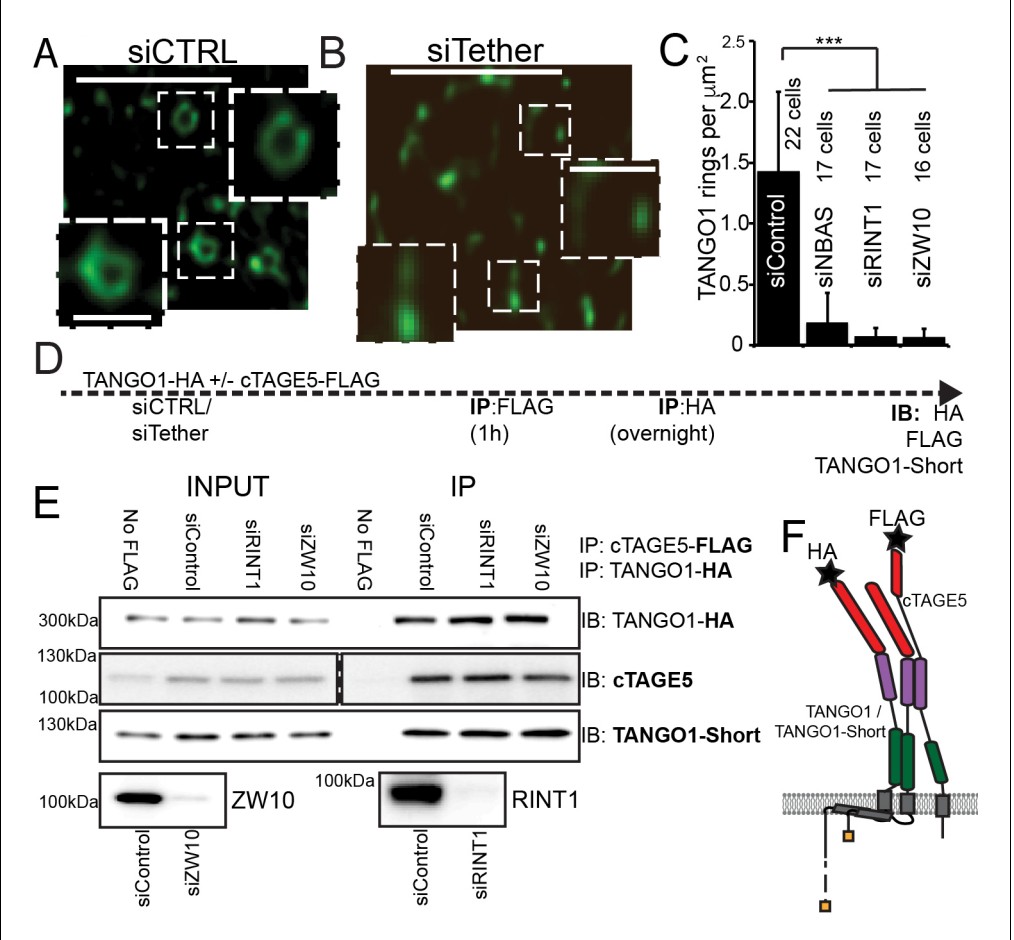

**Figure 7.** The NRZ tether is required for TANGO1 ring assembly siCTRL (**A**), siRINT1 (**B**), siNBAS, and siZW10 treated RDEB/FB/C7 were imaged by STED microscopy. TANGO1 rings in control cells (**A**). Representative image of a cell treated with siRINT1, showing almost no detectable assemblies of TANGO1 (**B**). The number of rings in each condition were manually counted and plotted (**C**) normalised to the area of collagen accumulations. The number of cells used in the quantification for each condition is indicated. (**D**) Schematic of experiment. Cells transfected with siRNA control, RINT1 or ZW10, lysed and subjected to sequential immunoprecipitations, (**E**). Eluates were probed for TANGO1-HA, cTAGE5, TANGO1-short. Cells with only TANGO1-HA (no cTAGE5-FLAG) were used as a negative control. Knockdown of RINT1 and ZW10 were confirmed by western blotting. (**F**) Schematic of a complex of TANGO1, TANGO1-short and cTAGE5 indicating positions of antibody epitopes used in the co-immunoprecipitations. Scale bars, (**A, B**) 1 μm, inset 400 nm. (**C**) ***p<0.001 (Student's t test).
DOI: https://doi.org/10.7554/eLife.32723.019

The following figure supplement is available for figure 7:

**Figure supplement 1.** ERES still form at collagen after depletion of tether proteins.
DOI: https://doi.org/10.7554/eLife.32723.020

remain at the nucleation site. This would explain the images presented (*Figure 6A* and *Figure 6—figure supplement 2*). Without the tethers, the reaction is stalled and TANGO1 fails to assemble further into a ring, providing an explanation for the requirement of tethers in TANGO1 ring assembly. An alternative is that the tethers are not recruited at the site of ring nucleation but present throughout, and we are unable to capture this final assembled state.

## TANGO1 as filament around COPII coat

We had proposed that TANGO1 functioned by binding to and stabilising the inner COPII coat to delay the recruitment of the outer coat and the subsequent fission of a newly forming carrier, for as long as is required to assemble and pack the bulky cargo collagen (*Saito et al., 2009*). We would

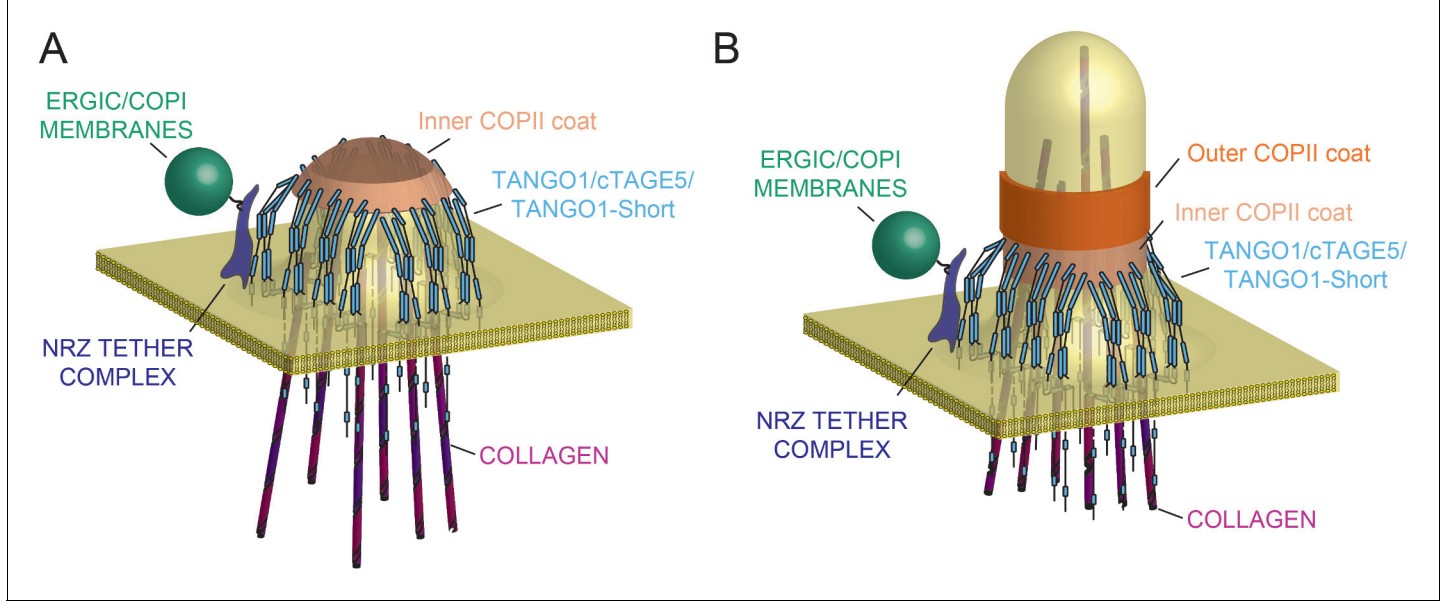

**Figure 8.** Model of TANGO1 ring assembly at an ERES. (**A**) TANGO1-family proteins (cyan) assembly into a ring at an ERES is mediated by interactions *1.* with COPII (orange) *2.* with triple helical collagen (purple), *3.* amongst the TANGO1 family proteins *4.* with the NRZ tether (dark blue) which links TANGO1 to ERGIC membranes. TANGO1 delays the binding of the outer COPII coat to allow a mega carrier to form. (**B**) The cytoplasmic bud grows to a size that encapsulates collagen trimers. In this form, we suggest that the neck of this tubule is covered in the inner COPII coat bound to TANGO1, which prevents premature recruitment of outer COPII coat, thereby controlling the timing of membrane fission.

DOI: https://doi.org/10.7554/eLife.32723.021

like to suggest a possible physical mechanism of how TANGO1 rings are assembled and maintained by means of protein-protein interactions and eventually regulate the formation of a collagen-containing megacarrier. First, based on our observations of TANGO1 rings by STED microscopy (*Raote et al., 2017*), (*Figure 2*), and our data indicating the different protein–protein interactions between the members of the TANGO1 family, we propose that a fence of TANGO1 can be described as a filament, held together by these lateral protein–protein interactions, which normally surrounds COPII patches at the ERES (*Maeda et al., 2017*; *Saito et al., 2011*). Importantly, this description of the ring as a filament will remain an approximation until the molecular composition and structural alignment of individual components is known. Such a filament would be subjected to elastic strains and stresses and would hence resist bending. Second, COPII subunits polymerise into structures of growing size. COPII subunits at the periphery of a polymerising domain have free binding sites and hence higher chemical energy than fully polymerised subunits at the centre of the domain, which, in physical terms, translates into the existence of an effective line-energy of the ERES. As proteins of the TANGO1 family physically interact with Sec23 (*Saito et al., 2009*; *Ma and Goldberg, 2016*; *Maeda et al., 2017*), Sec16 (*Maeda et al., 2017*), and Sec12 (*Saito et al., 2014*), we propose that upon adsorption to the ERES by binding peripheral COPII subunits, TANGO1 would effectively reduce the ERES line energy. A tug-of-war between the filament bending and the effect on COPII stabilisation created by the adsorption of TANGO1 filaments around ERES would then dictate whether and how TANGO1 rings are formed. Interestingly, it has been shown that the line tension of the polymerising protein coat can play a key role in controlling the timing and size of clathrin-coated vesicles (*Saleem et al., 2015*). We thus propose that the stabilising effect of TANGO1 while adsorbing around ERES would serve as a physical mechanism to delay and enlarge the COPII vesicle, commensurate with cargo size. Furthermore, TANGO1 rings could serve as a mould to impose a cylindrical curvature at the base of a growing carrier by coupling to the first layer of the COPII coat (*Figure 8A,B*), as proposed by Ma and Goldberg (*Ma and Goldberg, 2016*).

We expect that the diameter of a TANGO1 ring and associated components, will be maximal, proximal to the plane of the membrane. The more distal parts of the proteins for example the PRD (of TANGO1, cTAGE5 and TANGO1-short) will have two extreme positions:1, lying pointing radially

inward like spokes of a wheel and 2, pushed aside to the ring periphery by the growing carrier. It is therefore difficult to make definitive statements about relative locations - based on antibodies that bind to distal parts of the molecule - within the ring. We have also not tested whether cTAGE5, or for that matter TANGO1 short, can assemble into a ring in cells lacking TANGO1. We have not been able to create a form of cTAGE5 and TANGO1-short with a label or an antibody to visualise the domains proximal to the membranes, which makes it difficult to discern their location precisely, even in the presence of endogenous TANGO1. However, within these limitations, based on the involvement of various parts of TANGO1 and its interactors into discrete rings for collagen export, we could now begin to address the placement of various proteins such as TFG, KLHL12 or sedlin (*Johnson et al., 2015*; *McCaughey et al., 2016*; *Jin et al., 2012*; *Gorur et al., 2017*; *Venditti et al., 2012*) in collagen export from the ER.

Under these conditions, there is the possibility that a mega carrier, of the form recently reported by Schekman and colleagues (*Gorur et al., 2017*), is produced. Regardless of the final form adopted by the cells to transfer collagen from the lumen of the ER to the Golgi, with the data presented herein, we have taken the first steps toward arriving at a quantitative understanding of this hypothesis. We envision that a full description and analysis of such a quantitative physical model of TANGO1 ring assembly and megacarrier formation will help us better understand this fundamental process.

## TANGO1 links cargo folding to export

Little is understood about how client folding in the ER is coupled to export, how misfolded proteins and ER residents are excluded from an ERES, and what role the client plays in the biogenesis of its own carrier. TANGO1 recruits collagen via HSP47 – a chaperone that selectively recognises triple helical (export-competent) collagen (*Koide et al., 2000*; *Tasab et al., 2000*). Can this interaction of triple helical collagen and TANGO1 help effect ring assembly? Does a ring of TANGO1 (and therefore a carrier) form in response to selection of folded collagen, excluding misfolded collagen? Does folded cargo define the site and size of a transport carrier?

*In toto*, our data indicate that TANGO1, by assembling into a ring at ERES generates a semi-stable sub-domain across multiple compartments. The processes that allow this assembly also co-ordinately select, partition, and organise export machinery, and membrane for a cargo-export tubule/carrier, thus defining the minimal machinery for collagen export.

# Materials and methods

## Cell culture and transfection

RDEB/FB/C7, HEK293T and HeLa cells were grown at 37°C with 5% $CO_2$ in complete DMEM with 10% FBS unless otherwise stated. Plasmids were transfected in HeLa cells with TransIT-HeLa MONSTER (Mirus Bio LLC) or Lipofectamine 3000 Transfection Reagent (Thermo Fisher Scientific) according to the manufacturer's protocols. All cells in culture were tested every month to confirm they were clear of contamination by mycoplasma.

C-terminally HA-tagged full-length TANGO1 was cloned into the polylinker of pHRSIN transfer plasmid using BamHI/SalI restriction enzymes. Lentiviral particles were produced by co-transfecting pHRSIN-TANGO1-HA and a packaging vector pool (pCMV 8.91 and pMDG) into HEK293T cells using TransIT-293 (Mirus Bio LLC). 48 hr post transfection, the viral supernatant was harvested, filtered, and directly added to HEK293T cells. Stably expressing HEK293T cells were selected using 500 µg/ml hygromycin.

C-terminally FLAG-tagged full-length cTAGE5 was cloned into pJLM1 transfer plasmid using NheI/EcoRI restriction enzymes. Lentiviral particles were produced by co-transfecting pJLM1-cTAGE5-FLAG and a packaging vector pool (pPAX2 and pMD2.G) into HEK293T cells with using TransIT-293 (Mirus Bio LLC). 48 hr post transfection the viral supernatant was harvested, filtered, and directly added to TANGO1-HA expressing HEK293T cells. Cells stabling expressing Tango1-HA and cTAGE5-FLAG were selected using 500 µg/ml Hygromycin and Puromycin 4 µg/ml.

## Molecular biology

All molecular cloning was carried out using MAX Efficiency Stbl2 Competent Cells – (Thermo Fisher Scientific) following manufacturer's instructions.

### siRNAoligos

siRNA oligos were purchased from Eurofins Genomics (Ebersberg, Germany). The oligo sequences used were RINT1 5'-GGUUAUAACUGACAGGUAU-3', NBAS 5'-CUGCUUCAGUAUGGAUUAA ZW10 5'-UGGACGAUGAAGAGAAUUA-3', TANGO1 5'-GAUAAGGUCUUCCGUGCUU-3', cTAGE5 5'-UUGAAGACUCCAAAGUACA-3', SAR1A 5'-GAACAGAUGCAAUCAGUGATT-3', SAR1B 5'-GCA UAACUUGAAUUCAAUATT-3'. SEC23A siRNA (Cat # L-009582–01) was purchased from GE Dharmacon (Colorado, USA).

### Antibodies

The following antibodies were used collagen VII (rabbit anti–human [Abcam]; mouse anti–human [Sigma-Aldrich]), ERGIC-53 (mouse anti–human; Santa Cruz Biotechnology, Inc., and Enzo Life Sciences), Sec31A (mouse anti–human; BD), TANGO1 (rabbit anti–human; Sigma-Aldrich; rabbit anti–human in-house), HSP47 and calreticulin (goat anti–human; Enzo Life Sciences), HA (mouse; BioLegend), SAR1 (mouse anti–human; Abcam), β-tubulin (mouse anti-human; SIGMA-Aldrich), β-actin (mouse anti-human; SIGMA-Aldrich), NBAS (rabbit anti-human SIGMA-Aldrich), RINT1 (rabbit anti-human; SIGMA-Aldrich and goat anti-human (Santa Cruz Biotechnology), ZW10 (rabbit anti-human; Abcam), Sec23 (rabbit anti-human/mouse/rat; Abcam), cTAGE5 (rabbit anti-human Atlas antibodies, mouse anti-human Santa Cruz Biotechnology), TGN46 (sheep polyclonal, Bio-Rad), HA (mouse monoclonal, BioLegend; rat monoclonal BioLegend), FLAG (mouse monoclonal, rabbit, SIGMA-Aldrich; goat, Novus) HSP60 (mouse anti-human SIGMA-Aldrich), c-myc (mouse monoclonal, rabbit, SIGMA-Aldrich). Mounting media used in confocal and STED microscopy were either Vectashield (Vector Laboratories) or ProLong (Thermo Fisher Scientific, Waltham, Massachusetts).

### Immunoprecipitation and western blotting

Cells extracted with lysis buffer consisting of 50 mM Tris-HCl (pH 7.4), 150 mM NaCl, 1 mM EDTA, 2% CHAPS, and protease inhibitors were centrifuged at $20,000 \times g$ for 30 min at 4°C. Cell lysates were immunoprecipitated with FLAG M2 (SIGMA-Aldrich) or HA (Thermo Scientific) antibodies. Beads were washed three times with Tris-buffered saline (TBS)/0.5% CHAPS and processed for sample preparation.

For sequential immunoprecipitations, a first immunoprecipitation with FLAG would bring all proteins that interact with cTAGE5; a subsequent immunoprecipitation with HA would only yield proteins that were bound to both cTAGE5 and TANGO1-HA.

### Immunofluorescence microscopy

Cells grown on coverslips were fixed with cold methanol for 8 min at −20°C or 4% formaldehyde (Ted Pella, Inc.) for 15 min at room temperature. Cells fixed with formaldehyde were permeabilised with 0.1% Triton in PBS and then incubated with blocking reagent (Roche) or 0.1% horse serum for 30 min at room temperature. Primary antibodies were diluted in blocking reagent or 0.1% horse serum and incubated overnight at 4°C or at 37°C for 1 hr. Secondary antibodies conjugated with Alexa Fluor 594, 488, or 647 were diluted in blocking reagent and incubated for 1 hr at room temperature.

Confocal images were taken with a TCS SP5 (63×, 1.4–0.6 NA, oil, HCX PL APO), TCS SP8 (63×, 1.4 NA, oil, HC PL APO CS2), all from Leica Microsystems, using Leica acquisition software. Lasers and spectral detection bands were chosen for the optimal imaging of Alexa Fluor 488, 594, and 647 signals. Two-channel colocalisation analysis was performed using ImageJ (National Institutes of Health), and the Manders' correlation coefficient was calculated using the plugins JaCop or Coloc 2.

### STED microscopy

STED images were taken on a TCS SP8 STED 3 × microscope (Leica Microsystems) on a DMI8 stand using a 100 × 1.4 NA oil HCS2 PL APO objective and a pulsed supercontinuum light source (white light laser). Images were acquired and deconvolved exactly as described before (*Raote et al., 2017*).

Three-colour STED: Due to incompatible species specificities of primary antibodies available (for RINT1, TANGO1 and ERGIC-53), we were forced to use sub-optimal secondary antibodies. We used

Alexa 488, Alexa 594 and Alexa 647. This required that we set the depletion laser (775 nm) at only 3–8% intensity for the Alexa 647 channel to prevent rapid bleaching.

## Morphology quantification of TANGO1 rings

Multichannel 3D stacks were acquired with a z-step size of 100 nm and subsequently deconvolved using Huygens deconvolution software (Scientific Volume Imaging) for STED modes using shift correction to account for drift during stack acquisition. Sum-Intensity Projections were then generated from a subset of the deconvolved stack slices where the rings were present. Projected images showed a large fraction of the GFP signal as random dots or big aggregates in which no particular structural organisation could be distinguished. Also, a significant amount of well-defined non-random structures, i.e. both full and incomplete (arc-shaped or dotted) rings, as well as chain-like assemblies of rings.

To ensure a systematic and unbiased analysis, these structures are first segmented via a trainable pixel level classifier, and subsequently labelled either as rings, incomplete rings or dots, or ring aggregates, on object level. Both pixel and object classification used a machine learning based open-source software, ilastik (*Sommer et al., 2011*). Afterwards, we calculated different parameters for each object to compare them quantitatively in shape and size. Specifically, we measured the diameters of the ring in terms of major and minor axes of its fitted ellipse and the maximum and minimum Feret's diameter. Statistical testing was performed using Student's t test (continuous data, two groups). One asterisk indicates Student's t test value $p<0.06$; three asterisks $p<0.006$; ns indicates not significant.

To quantify the frequency of rings after depletion of specific gene products, deconvolved STED images of each condition were manually scored for rings/clusters of TANGO1. A ring is defined as an independent structure with an internal hole. A cluster however, is at least four such conjoint rings. Statistical testing was performed using Student's t test (continuous data, two groups). One asterisk indicates Student's t test value $p<0.02$; three asterisks $p<0.002$; ns indicates not significant.

## Collagen secretion assay in RDEB/FB/C7 fibroblasts

The secretion assay was carried out exactly as described earlier (*Nogueira et al., 2014*; *Santos et al., 2015*). Briefly, RDEB/FB/C7 fibroblasts were transfected in suspension on two consecutive days with siRNA (either a pool of control, non-targeting RNA or RNA targeting a specific gene). 48 hr later, cells were washed thoroughly and incubated for 20 hr in OptiMEM supplemented with 1 mM ascorbate. Cell lysates and media were harvested and processed for Western blotting of collagen VII and tubulin/actin as loading/lysis controls.

## Acknowledgements

We thank the Advanced Light Microscopy Unit at the CRG, Javier Diego Iñiguez, Verena Ruprecht and members of the Malhotra laboratory for valuable discussions. V Malhotra is an Institució Catalana de Recerca i Estudis Avançats professor at the Centre for Genomic Regulation, the work in his laboratory is funded by grants from the Ministerio de Economía, Industria y Competitividad Plan Nacional (ref. BFU2013-44188-P) and Consolider (CSD2009-00016). We acknowledge support of the Spanish Ministry of Economy and Competitiveness, through the Programmes 'Centro de Excelencia Severo Ochoa 2013–2017' (SEV-2012–0208) and Maria de Maeztu Units of Excellence in R and D (MDM-2015–0502). We acknowledge the support of the CERCA Programme/Generalitat de Catalunya. F Campelo and M García-Parajo acknowledge support by the Spanish Ministry of Economy and Competitiveness ('Severo Ochoa' Programme for Centres of Excellence in R and D (SEV-2015–240522) and FIS2014-56107-R), BFU2015-73288-JIN, AEI/FEDER; UE, Fundacion Privada Cellex, HFSP (GA RGP0027/2012), EC FP7-NANO-VISTA (GA 288263) and LaserLab 4 Europe (GA 654148). I. Raote and F. Campelo acknowledge support by the BIST Ignite Grant (eTANGO). This work reflects only the authors' views, and the EU Community is not liable for any use that may be made of the information contained therein.

# Additional information

## Competing interests

Vivek Malhotra: Senior editor, *eLife*. The other authors declare that no competing interests exist.

## Funding

| Funder | Grant reference number | Author |
|---|---|---|
| Ministerio de Economía y Competitividad | BFU2013-44188-P | Vivek Malhotra |
| Ministerio de Economía y Competitividad | CSD2009-00016 | Vivek Malhotra |
| Barcelona Institute of Science and Technology | BIST-IGNITE-eTANGO | Ishier Raote Felix Campelo Vivek Malhotra |
| Ministerio de Economía y Competitividad | SEV-2012-0208 | Vivek Malhotra |
| Ministerio de Economía y Competitividad | SEV-2015-240522 | Vivek Malhotra |
| Ministerio de Economía y Competitividad | FIS2014-56107-R | Maria F Garcia-Parajo |
| Ministerio de Economía y Competitividad | MDM-2015-0502 | Vivek Malhotra |
| Ministerio de Economía y Competitividad | BFU2015-73288-JIN | Maria F Garcia-Parajo |
| Fundacion Privada Cellex | | Maria F Garcia-Parajo |
| AEI/FEDER, UE | | Maria F Garcia-Parajo |
| Human Frontier Science Program | GA RGP0027/2012 | Maria F Garcia-Parajo |
| Cordis | EC FP7-NANO-VISTA (GA 288263) | Maria F Garcia-Parajo |
| LaserLab 4 Europe | GA 654148 | Maria F Garcia-Parajo |

The funders had no role in study design, data collection and interpretation, or the decision to submit the work for publication.

## Author contributions

Ishier Raote, Conceptualization, Data curation, Formal analysis, Supervision, Funding acquisition, Validation, Investigation, Visualization, Methodology, Writing—original draft, Project administration, Writing—review and editing; Maria Ortega-Bellido, Validation, Investigation, Methodology, Project administration; António JM Santos, Formal analysis, Validation, Investigation, Visualization; Ombretta Foresti, Resources, Investigation, Writing—review and editing; Chong Zhang, Resources, Software, Formal analysis, Visualization; Maria F Garcia-Parajo, Supervision, Funding acquisition; Felix Campelo, Conceptualization, Resources, Formal analysis, Supervision, Funding acquisition, Investigation, Visualization, Methodology, Writing—original draft, Project administration, Writing—review and editing; Vivek Malhotra, Conceptualization, Supervision, Funding acquisition, Writing—original draft, Project administration, Writing—review and editing

## Author ORCIDs

Ishier Raote http://orcid.org/0000-0002-5898-4896
Ombretta Foresti http://orcid.org/0000-0002-6878-0395
Felix Campelo http://orcid.org/0000-0002-0786-9548
Vivek Malhotra http://orcid.org/0000-0001-6198-7943

Decision letter and Author response
Decision letter https://doi.org/10.7554/eLife.32723.024
Author response https://doi.org/10.7554/eLife.32723.025

## Additional files

### Supplementary files

• Transparent reporting form
DOI: https://doi.org/10.7554/eLife.32723.022

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
