## [Decision Letter]

Thank you for submitting your article "Building a machine for collagen export from the endoplasmic reticulum" for consideration by *eLife*. Your article has been favorably evaluated by Ivan Dikic (Senior Editor) and three reviewers, one of whom is a member of our Board of Reviewing Editors. The reviewers have opted to remain anonymous.

The reviewers have discussed the reviews with one another and the Reviewing Editor has drafted this decision to help you prepare a revised submission.

Overall, the reviewers agree that the manuscript will require a careful and thorough revision that omits the model unless it can be tested experimentally. In addition, they all felt that each of the phenotypes would benefit from much better documentation, clarification and quantitation. Although we normally delete the individual reviews, we include them here to guide your revision efforts. We hope you will find these comments constructive in preparing a revised manuscript.

*Reviewer #1:*

TANGO1 interacts with CTAGE5 and COPII components Sec23/Sec24 and recruits ERGIC-53 containing membranes to generate a mega-transport carrier for export of collagens from the ER. Malhotra and colleagues recently showed show TANGO1 assembles into a ring that encircles COPII components. They have shown that growth of transport carriers for bulky cargoes requires addition of membranes; TANGO1 remains at the neck of the newly forming transport carrier, which grows in size by addition of ERGIC-53-containing membranes to generate a transport intermediate for collagen export.

In this paper, the authors explore the domain requirements for RING formation and explore a computational model for how these structures grow and form. First they find that removal of TANGO1's proline rich region (PRD) changes the structures formed in cells to smaller rings (is it clear they are not small patches?) or interesting long linear assemblies (Figure 2N not K). This suggests that interaction with Sec23 redirects a self-assembly process into a productive carrier formation event. This leads the authors to propose a model whereby free ends have more "energy" than occupied subunits, creating "line energy"; they propose that TANGO1 proteins act as lineactants upon binding Sec23 (they showed that Sec23 availability alters the polymerization of TANGO proteins.) In compelling STED images, they add new data showing a role for the NRZ tether in collagen secretion, linked to TANGO.

Overall, there is important and interesting data here but the story is not told in an adequately documented and coherent manner. This reviewer feels that the roles of domains should be better documented for each of the constructs (more images, better quantitation) and the theoretical lineactant model presented elsewhere as it does not add to the present story and should be based on actual data for filament stiffness etc. Better statistics are needed to discern the probability of colocalization of the tether with ERGIC and TANGO1 (and to substantiate all the structural forms). Also, care should be taken in analyzing Sec23 knockdowns as cells (and ER) will not be normal. Finally, the paper would be greatly enhanced by EM analysis of purified protein constructs but this is likely not yet possible.

The authors propose a model whereby "filament resistance to bending leads to ERES wetting or dewetting by TANGO1 filaments." This is one of many possibilities. Models are valuable when they can be tested directly, and the model seems far ahead of the available data presented. At this point, it does not add to the story, and should be presented with testable predictions to support it. Thus, this reviewer feels strongly that the model should be presented elsewhere until tests can be included. Also, the authors propose a model for ring fusion with no data that they actually fuse.

Many references to figure legends are not correct and legends should be made clearer; please include page numbers.

Figure 8 should include what happens after long collagen exceeds length of TANGO.

Number of cells counted; number of particles scored are both missing throughout.

*Reviewer #2:*

This paper continues the investigation by the Malhotra lab into the ER export of bulky cargoes such as collagen, and the role of TANGO1 in this process. The goal is to understand how TANGO1 forms a ring of a specific size around COPII at ER exit sites (ERES). A new role for the NRZ (NBAS/RINT1/ZW10) tether in generating TANGO1 rings and linking TANGO1 to ERGIC membranes is described. These data are integrated into a model for the "collagen export machine" at the ER.

The domain structure of TANGO1, and the functions of the individual domains, are now fairly well understood. This information is exploited here by deleting individual domains to test how TANGO1 assembles into rings. Deletion of the COPII-binding proline-rich domain generates fused or linear assemblies of rings. This observation is interpreted in terms of a rather elaborate theoretical model, which assumes that TANGO1 and its partners can form linear filaments that normally interact with COPII. The conclusion is that TANGO1 acts as a "lineactant" at ERES. I was unfamiliar with this term, but it implies that TANGO1 acts to reduce line tension at the edges of ERES.

This analysis is intriguing and may well have validity, but it would benefit from direct evidence that TANGO1 and its partners have the intrinsic ability to form filaments. While the experiments described here are useful for elucidating the roles of TANGO1 domains and partner proteins in ring formation, they do not directly show that filaments are present. It would also be nice if the model made a testable and nontrivial prediction of a phenomenon that had not already been observed. Despite these reservations, the modeling seems to have been useful for guiding the structure-function analysis.

Other points from the experimental data:

1) I'm not clear on the significance of Figure 3, which shows that deleting the luminal SH3 domain disrupts TANGO1 rings. Was that result expected? If so, why? The explanation in the text about elastic properties of the TANGO1 filament lacks a clear justification.

2) In Figure 4, cTAGE5 associates with TANGO1 rings, but the two signals do not overlap. If these proteins co-polymerize in a filament, shouldn't they have very similar distributions?

3) Figure 5 is confusing. I can't see any difference between TRUNC1 and TRUNC2 in Figure 5. In Figure 5, aren't TRUNC1 and TRUNC2 reversed? The text says that TRUNC2 showed no recruitment, but it gives the higher Manders' coefficient. Also, the text seems to have errors in referring to Figure 5 and a nonexistent Figure 5G.

4) For Figure 7, are ERES still present when the NRZ tether is depleted? In other words, is the effect on TANGO1 rings direct or indirect? The interactions described here between TANGO1 and the NRZ tether seem to be real, but the mechanistic interpretation remains to be clarified.

My overall impression is that this manuscript is ambitious and interesting. It provides a substantial amount of new data about TANGO1 domain functions and protein-protein interactions at ERES. The ideas are stimulating, but they go further beyond the data than may be prudent.

*Reviewer #3:*

The authors continue their quest to understand how large cargoes are accommodated in COPII vesicles. In a recent JCB paper, they reported that TANGO1, known to be important for collagen export from the ER, assembles into rings around COPII coats at ER exit sites. Here they use high-resolution imaging and deletion mutagenesis to further characterize lateral interactions that stabilize the rings, including a potential role for the not-too-well-characterized NRZ tethering complex. They also use modeling to generate a theoretical phase diagram for ring formation and fusion (my first encounter with the word "lineactant", which according to a bit of Googling is rather obscure). I'm on the fence about whether this work represents a sufficient advance for *eLife*.

I think the most novel, and potentially most interesting, finding is that the NRZ tether complex is directly involved in TANGO ring formation. Knocking down RINT-1, however, strikes me as a blunt instrument for testing this hypothesis – how can we know that the effect on ring formation (which in any case is not quantified) is direct, given the central role of this complex in trafficking within the early secretory pathway? I wonder if the authors also have their doubts, since a direct role for the NRZ complex in ring formation is not depicted in Figure 8.

I'm not, unfortunately, qualified to critique the modeling component of this manuscript. Nonetheless, I am not convinced of the wisdom of combining in vivo experiments like the ones presented here – rather drastic manipulations in a complex milieu – with abstract mathematical modeling, absent some intermediate in vitro measurements on model membranes using defined protein components. The authors tout the success of the model in predicting that reducing the interaction energy between TANGO1 filaments and COPII subunits enhances ring fusion. Yet it seems to me that there is a 50:50 chance that any such prediction would prove correct by random chance – after all, ring fusion is bound to be either more efficient or less efficient. Quantitative agreement over a range of experimental conditions would support the modeling work much more compellingly. Without it, I wonder if the modeling belongs.

None of the explanations offered for the smaller rings observed in vivo for TANGO1deltaPRD strike me as likely. For example, why (if the domains are independent, which is the rationale behind the whole approach) would deleting PRD affect the bending rigidity or preferred curvature of the filament?

Maybe I missed it, but I didn't understand what the authors take to be the mechanistic implications of the finding that "the minimal TEER is exactly the same forty amino acids we identified in the previous section, as those required for the self-association of TANGO1. This had clear implications for the coupling of the tethering function of TANGO1 with its structure, homo-oligomerisation and thereafter, assembly into rings." What are the implications?

I had a lot of trouble with the figures, ranging from figures that weren't there (Figure 3—figure supplement 2) to figures that were bizarrely laid out (Figure 2 is really quite extraordinary, with font sizes that must differ by an order of magnitude and panel labels in every imaginable place) to Figure 4—figure supplement 2 which, in my view, represents modeling malpractice.

Finally, what could be the purpose of including line numbers while omitting page numbers?

[Editors' note: further revisions were requested prior to acceptance, as described below.]

Thank you for submitting your article "Building a machine for collagen export from the endoplasmic reticulum" for consideration by *eLife*. Your article has been favorably evaluated by Ivan Dikic (Senior Editor) and three reviewers, one of whom, Suzanne Pfeffer, is a member of our Board of Reviewing Editors.

The reviewers have discussed the reviews with one another and the Reviewing Editor has drafted this decision to help you prepare a revised submission.

One of the reviewers wrote, "The mechanistic picture still seems to be clearer to the authors than to me as a reader, but this work is a significant step in the right direction." Given this concern, the following edits are offered to improve the story so it will be best appreciated by *eLife* readers.

TANGO1 interacts with CTAGE5 and COPII components and recruits ERGIC-53 membranes to generate a mega-transport carrier for export of collagens from the ER. Malhotra recently showed in JCB that TANGO1 assembles into a ring at the ER that encircles COPII components. Here they go on to show that ring formation requires Sec23, self-interaction and interaction with cTAGE5, and the presence of the RINT tethering complex.

In general the work will be of broad interest. However, the manuscript could benefit from careful rewriting in a few places, as the reviewers had a difficult time parsing all the abbreviations, mutations, and relationship to multiple isoforms of TANGO proteins. The following presentation improvements will enhance the ability of a reader to understand the story.

1) Perhaps the most important new and unexpected finding shown here is the recruitment of the RINT complex to the TANGO rings. The RINT forms a clear focus of staining on one side of the ring, yet seems essential for RING assembly. What fraction of TANGO rings showed RINT association? How many were counted? The authors should also present categorizations of where the RINT labeled the rings – (inside? at edge? outside?) The authors include hand-waving explanations for why a full RINT ring may not have been seen but should really think about how a focus of RINT could nevertheless stabilize a RING structure and discuss this thoughtfully.

2) Figure 1. It would help the reader to indicate that the cytoplasmically oriented PRD is at the C-terminus here (indicate -COOH and include parenthesis PRD). So many words detract from the clarity. Could the authors use arrows from interacting proteins to the indicated domains? If function is unknown, it is not necessary to state this in the diagram. References can go in the legend. Entire figure could be 1/4 the size. The linear version at the bottom should be removed; Figure 2 should be shown at the top of Figure 1 as TEER domain is not mentioned in diagram.

3) It was very hard for this reader to be able to compare wild type and mutant form rings because they are all shown at slightly different magnifications. Figure 2, should be presented at the same mag. Same for Figure 3; Figure 4; In any frame in which the mag bar is 20µm, please outline the cell. In Figure 2, what is the difference between C, D and E? In general, if the figure is very crowded, please consider multiple figures to make the beautiful images be best appreciated by *eLife* readers.

4) Does cTAGE5 make rings in cells lacking TANGO1? Please summarize what the data reveal about roles of TANGO versus TANGO short (please define!) versus cTAGE5.

5) Linear filaments in a planar membrane could generate rings, but association of the rings with one another must require multivalent interactions that go beyond head-to-tail filament formation. This point should be mentioned.

6) The authors propose that "TANGO1/cTAGE5 interactions with COPII […] act as an inward attractant, holding the filament in a ring-like configuration." But then why would the δ-PRD mutant still form rings? Perhaps they mean to state that the COPII interaction constrains the size and placement of the rings.

7) The minimal TEER is the same 40 amino acids that are required for TANGO1 self-association. What does this mean? A crucial point that should be addressed is whether these two interactions are mutually exclusive.

8) Subsection “Compartment tethering in a TANGO1 ring assembly pathway”, tenth paragraph: the text refers to quantitation in Figure 6, but should it be Figure 7?

9) "In all cases, exit sites (as marked by TANGO1 and SEC31) are still recruited to intracellular collagen accumulations (Figure 7—figure supplement 1)." I don't see that result in Figure 7—figure supplement 1, and actually, I'm not sure what that supplemental figure is intended to show.

10) Abstract: "for"? its self-association?

Subsection “Binding of TANGO1 to COPII controls TANGO1 ring formation”, second paragraph: TANGO1-short is not defined anywhere.

"CTAGE is on the periphery of rings." How many rings were counted to give confidence to this conclusion? How was this quantified?

"linear attraction along the length of the filament". By filament do they mean along the length of TANGO1? The authors should add a sentence stating that the interactions are *consistent* with. They have not defined the actual structural relationships here.

"has the potential to segregate cargo". This has not yet been shown.

---

## [Author Response]

Overall, the reviewers agree that the manuscript will require a careful and thorough revision that omits the model unless it can be tested experimentally.

With a heavy heart, we remove the model. The model gave us a potential way to propose the physical nature of the events leading to the assembly of TANGO1 into a ring. Our biophysical colleagues saw this as a major step forward. Admittedly, we haven’t the means to test some of its features experimentally. So, the model is out.

In addition, they all felt that each of the phenotypes would benefit from much better documentation, clarification and quantitation.

We have now gone through the paper and extensively revised the text to better document our procedures and data. We have quantified all that was possible, which includes effects on ring formation after depleting the several specific gene products.

Although we normally delete the individual reviews, we include them here to guide your revision efforts. We hope you will find these comments constructive in preparing a revised manuscript.

Our specific comments to the reviewers follow.

Reviewer #1:[…] In this paper, the authors explore the domain requirements for RING formation and explore a computational model for how these structures grow and form. First they find that removal of TANGO1's proline rich region (PRD) changes the structures formed in cells to smaller rings (is it clear they are not small patches?) or interesting long linear assemblies (Figure 2N not K). This suggests that interaction with Sec23 redirects a self-assembly process into a productive carrier formation event. This leads the authors to propose a model whereby free ends have more "energy" than occupied subunits, creating "line energy"; they propose that TANGO1 proteins act as lineactants upon binding Sec23 (they showed that Sec23 availability alters the polymerization of TANGO proteins.) In compelling STED images, they add new data showing a role for the NRZ tether in collagen secretion, linked to TANGO.Overall, there is important and interesting data here but the story is not told in an adequately documented and coherent manner. This reviewer feels that the roles of domains should be better documented for each of the constructs (more images, better quantitation) and the theoretical lineactant model presented elsewhere as it does not add to the present story and should be based on actual data for filament stiffness etc.

We have removed the model. We have included additional images for each construct. We have described in detail our scheme of quantification of shape and size and rings. We specifically mention the number of objects/rings and cells analysed. We have quantified the effect on the frequency of rings after depleting cells of specific gene products. Many of the images are presented differently, as suggested by the reviewer.

Better statistics are needed to discern the probability of colocalization of the tether with ERGIC and TANGO1 (and to substantiate all the structural forms).

The colocalisation of tethers to TANGO1 and Sec31 is now presented.

Also, care should be taken in analyzing Sec23 knockdowns as cells (and ER) will not be normal.

We have depleted only Sec23A, which would leave Sec23B, and therefore attempt to limit cellular endomembrane stress. There is simply no other way to address the involvement of COPII without reducing their levels. But based on this concern, we have now included a statement in the text to frame our findings with this caveat.

Finally, the paper would be greatly enhanced by EM analysis of purified protein constructs but this is likely not yet possible.

Yes, we would like to do this and we are trying to convince Wolfgang Baumeister to help visualise this arrangement of TANGO1 by cryo-electron microscopy. But this will take a long time.

The authors propose a model whereby "filament resistance to bending leads to ERES wetting or dewetting by TANGO1 filaments." This is one of many possibilities. Models are valuable when they can be tested directly, and the model seems far ahead of the available data presented. At this point, it does not add to the story, and should be presented with testable predictions to support it. Thus, this reviewer feels strongly that the model should be presented elsewhere until tests can be included.

We have removed the model.

Also, the authors propose a model for ring fusion with no data that they actually fuse.

We agree that we based on our data we cannot state that the rings actually fuse together, since the only way to test this would be by reconstitution and cryo-electron microscopy. However, the TANGO1 rings appear structurally different to independent rings, a structure morphologically resembling that of fused rings. We have rephrased our words to more precisely describe our observations and their interpretation.

Many references to figure legends are not correct and legends should be made clearer; please include page numbers.

Done.

Figure 8 should include what happens after long collagen exceeds length of TANGO.

Our data is all about the assembly of a ring of TANGO1 at the neck for the newly forming collagen-containing carriers. Whether it grows and is released in the form of a mega carrier or, for example, fuses directly to the Golgi cannot be said at all. But, we now included in Figure 8 what we propose is the next step in the growth of the container as a large tubule where the tip is uncoated and the based containing the coats is captured by TANGO1.

Number of cells counted; number of particles scored are both missing throughout.

Done.

Reviewer #2:[…] This analysis is intriguing and may well have validity, but it would benefit from direct evidence that TANGO1 and its partners have the intrinsic ability to form filaments. While the experiments described here are useful for elucidating the roles of TANGO1 domains and partner proteins in ring formation, they do not directly show that filaments are present. It would also be nice if the model made a testable and nontrivial prediction of a phenomenon that had not already been observed. Despite these reservations, the modeling seems to have been useful for guiding the structure-function analysis.

As mentioned above, we have removed the model to alleviate the concerns raised here about our inability to test this directly. It is a pity, because it did guide the structure-function analysis, as the reviewer points out.

Other points from the experimental data:1) I'm not clear on the significance of Figure 3, which shows that deleting the luminal SH3 domain disrupts TANGO1 rings. Was that result expected? If so, why? The explanation in the text about elastic properties of the TANGO1 filament lacks a clear justification.

We have removed the SH3 domain data. This could have made sense in light of the physical model, but now it seems a bit odd to be included in the text.

2) In Figure 4, cTAGE5 associates with TANGO1 rings, but the two signals do not overlap. If these proteins co-polymerize in a filament, shouldn't they have very similar distributions?

It is very hard to capture these images with STED because of the quality of reagents, the plane of the ER, and the spatial distance between antibody epitopes in these proteins. However, we now present other images where c-TAGE5 also appears as part of the TANGO1 rings.

3) Figure 5 is confusing. I can't see any difference between TRUNC1 and TRUNC2 in Figure 5. In Figure 5, aren't TRUNC1 and TRUNC2 reversed? The text says that TRUNC2 showed no recruitment, but it gives the higher Manders' coefficient. Also, the text seems to have errors in referring to Figure 5 and a nonexistent Figure 5G.

Thank you for pointing this out. This is now corrected.

4) For Figure 7, are ERES still present when the NRZ tether is depleted? In other words, is the effect on TANGO1 rings direct or indirect? The interactions described here between TANGO1 and the NRZ tether seem to be real, but the mechanistic interpretation remains to be clarified.

We now show that ERES are still present under our experimental conditions.

My overall impression is that this manuscript is ambitious and interesting. It provides a substantial amount of new data about TANGO1 domain functions and protein-protein interactions at ERES. The ideas are stimulating, but they go further beyond the data than may be prudent.

We thank the reviewer for noting our ambitious description of the interesting data. We are not trying to over sell the goods, but we believe this is a new discovery and a reasonable explanation of how large cargoes are collected and exported.

Reviewer #3:The authors continue their quest to understand how large cargoes are accommodated in COPII vesicles. In a recent JCB paper, they reported that TANGO1, known to be important for collagen export from the ER, assembles into rings around COPII coats at ER exit sites. Here they use high-resolution imaging and deletion mutagenesis to further characterize lateral interactions that stabilize the rings, including a potential role for the not-too-well-characterized NRZ tethering complex. They also use modeling to generate a theoretical phase diagram for ring formation and fusion (my first encounter with the word "lineactant", which according to a bit of Googling is rather obscure).

The concept of *lineactant* or *line active* agents has been studied in soft matter physics and biophysics in the context of lipid domain formation (see e.g. the works on line active lipids by Sam Safran or Siewert Jan Marrink or Marcus Müller). One of us (Malhotra) also had to spend considerable time in trying to understand the principle of lineactant. But, as mentioned above, our biophysicists friends started the evaluation of our data by summarizing it as a lineactant.

I'm on the fence about whether this work represents a sufficient advance for eLife.

We hope that given the current understanding of how cargoes, especially of the bulky and complicated kind, are exported form the ERES, and our data on how TANGO1 interacts with components of COPII and ERGIC via the tethers, we are helping to illuminate this rather dark spot in membrane traffic. We sincerely hope that the kind reviewer is persuaded to side with us.

I think the most novel, and potentially most interesting, finding is that the NRZ tether complex is directly involved in TANGO ring formation. Knocking down RINT-1, however, strikes me as a blunt instrument for testing this hypothesis – how can we know that the effect on ring formation (which in any case is not quantified) is direct, given the central role of this complex in trafficking within the early secretory pathway?

We have now tested each of the three proteins of the tether complex individually and describe quantitatively, that all proteins of this complex have an equal effect on ring formation.

I wonder if the authors also have their doubts, since a direct role for the NRZ complex in ring formation is not depicted in Figure 8.

The reason for not showing NRZ in a ring of TANGO1 is because we do not detect these proteins in a ring. It remains possible that these interactions are transient and will become evident in live cell super resolution imaging, which is currently impossible. But we appreciate the reviewers concern and have included this caveat in the Discussion.

I'm not, unfortunately, qualified to critique the modeling component of this manuscript. Nonetheless, I am not convinced of the wisdom of combining in vivo experiments like the ones presented here – rather drastic manipulations in a complex milieu – with abstract mathematical modeling, absent some intermediate in vitro measurements on model membranes using defined protein components. The authors tout the success of the model in predicting that reducing the interaction energy between TANGO1 filaments and COPII subunits enhances ring fusion. Yet it seems to me that there is a 50:50 chance that any such prediction would prove correct by random chance – after all, ring fusion is bound to be either more efficient or less efficient. Quantitative agreement over a range of experimental conditions would support the modeling work much more compellingly. Without it, I wonder if the modeling belongs.

Sadly, we have removed the model.

None of the explanations offered for the smaller rings observed in vivo for TANGO1deltaPRD strike me as likely. For example, why (if the domains are independent, which is the rationale behind the whole approach) would deleting PRD affect the bending rigidity or preferred curvature of the filament?

We have removed the physical model, and thereby no longer make this explanation. However, it is apparent that we were unable to provide a clear explanation of our model's results, as we agree entirely with the reviewer in thinking that the intrinsic bending rigidity and preferred curvature of a filament should remain unaffected. Yet the rings are smaller, which we explained based on the results of our model, by the fact that TANGO1 proline rich domain, by binding to Sec23 and Sec16, has the ability to modify normal ERES dynamics (included in the model as the coupling parameter f_0) (see e.g. Figure 3 or Appendix Figure 1H in our initial submission).

Maybe I missed it, but I didn't understand what the authors take to be the mechanistic implications of the finding that "the minimal TEER is exactly the same forty amino acids we identified in the previous section, as those required for the self-association of TANGO1. This had clear implications for the coupling of the tethering function of TANGO1 with its structure, homo-oligomerisation and thereafter, assembly into rings." What are the implications?I had a lot of trouble with the figures, ranging from figures that weren't there (Figure 3—figure supplement 2) to figures that were bizarrely laid out (Figure 2 is really quite extraordinary, with font sizes that must differ by an order of magnitude and panel labels in every imaginable place) to Figure 4—figure supplement 2 which, in my view, represents modeling malpractice.Finally, what could be the purpose of including line numbers while omitting page numbers?

We have removed the mention of possible implications of the relation between the tether-binding site and the self-association motif. We have revised our text and images to improve layouts and readability. We have removed any mention of the ab initio structure prediction (Figure 4—figure supplement 2), which we had only used as a guide in designing deletions of a coiled coil domain.

[Editors' note: further revisions were requested prior to acceptance, as described below.]

One of the reviewers wrote, "The mechanistic picture still seems to be clearer to the authors than to me as a reader, but this work is a significant step in the right direction." Given this concern, the following edits are offered to improve the story so it will be best appreciated by eLife readers.TANGO1 interacts with CTAGE5 and COPII components and recruits ERGIC-53 membranes to generate a mega-transport carrier for export of collagens from the ER. Malhotra recently showed in JCB that TANGO1 assembles into a ring at the ER that encircles COPII components. Here they go on to show that ring formation requires Sec23, self-interaction and interaction with cTAGE5, and the presence of the RINT tethering complex.In general the work will be of broad interest. However, the manuscript could benefit from careful rewriting in a few places, as the reviewers had a difficult time parsing all the abbreviations, mutations, and relationship to multiple isoforms of TANGO proteins. The following presentation improvements will enhance the ability of a reader to understand the story.1) Perhaps the most important new and unexpected finding shown here is the recruitment of the RINT complex to the TANGO rings.

We appreciate this sentiment, but it is not just about the tethers. We too are very excited about the involvement of the tethers, but this paper is also about how various parts of TANGO1 and associates, including the tethers, assemble into higher order structures.

The RINT forms a clear focus of staining on one side of the ring, yet seems essential for RING assembly. What fraction of TANGO rings showed RINT association? How many were counted? The authors should also present categorizations of where the RINT labeled the rings – (inside? at edge? outside?) The authors include hand-waving explanations for why a full RINT ring may not have been seen but should really think about how a focus of RINT could nevertheless stabilize a RING structure and discuss this thoughtfully.

The numbers are now included in the text. We also include the following in the Discussion to address the reviewers concerns about the location of RINT1 with respect to TANGO1 ring.

“We have not observed a complete ring of tethers with TANGO1. The tethers instead appear as one or two puncta at the ring circumference. […] An alternative is that the tethers are not recruited at the site of ring nucleation but present throughout and we are unable to capture this final assembled state.”

“We expect that the diameter of a TANGO1 ring and associated components, will be maximal proximal to the plane of the membrane. […] However, within these limitations, based on the involvement of various parts of TANGO1 and its interactors into discrete rings for collagen export we could now begin to address the placement of various proteins such as TFG, KLHL12 or sedlin (Johnson et al., 2015; McCaughey et al., 2016; Jin et al., 2012; Gorur et al., 2017; Venditti et al., 2012) in collagen export from the ER.”

“Deconvolved z-stacks of images of ten cells were used to quantify the location of the tether protein RINT1 relative to a ring of TANGO1. 90 rings of TANGO1 were manually scored, three adjacent slices in the image stack were used to identify signal from RINT1 in the vicinity of the ring of TANGO1. 23 rings showed RINT1 within the ring, 19 rings showed RINT1 on the circumference (at the edge) of the ring, 39 had RINT1 outside the ring, 9 rings showed no detectable RINT1.” This has been included in the figure legend of Figure 6.

We also now show 14 panels of TANGO1 rings and RINT1 location in Figure 6—figure supplement 2.

2) Figure 1. It would help the reader to indicate that the cytoplasmically oriented PRD is at the C-terminus here (indicate -COOH and include parenthesis PRD). So many words detract from the clarity. Could the authors use arrows from interacting proteins to the indicated domains? If function is unknown, it is not necessary to state this in the diagram. References can go in the legend. Entire figure could be 1/4 the size. The linear version at the bottom should be removed; Figure 2 should be shown at the top of Figure 1 as TEER domain is not mentioned in diagram.

Done.

3) It was very hard for this reader to be able to compare wild type and mutant form rings because they are all shown at slightly different magnifications. Figure 2, should be presented at the same mag. Same for Figure 3; Figure 4; In any frame in which the mag bar is 20µm, please outline the cell. In Figure 2, what is the difference between C, D and E? In general, if the figure is very crowded, please consider multiple figures to make the beautiful images be best appreciated by eLife readers.

Done. Unfortunately, we cannot include the outline of cells, as we did not acquire a phase contrast image at the time.

4) Does cTAGE5 make rings in cells lacking TANGO1? Please summarize what the data reveal about roles of TANGO versus TANGO short (please define!) versus cTAGE5.

The Introduction now contains the following text.

Figure 1 is a schematic of three TANGO1 family proteins: TANGO1, TANGO1-short and cTAGE5. […] How different binding partners could affect the overall function of these proteins in ERES assembly and cargo export remains untested.”

The legend to Figure 1 now contains the following text.

“Figure 1. The domain architecture and topology of TANGO1 and cTAGE5

(A) A schematic depiction of full length TANGO1, showing the extent of each domain in amino acids. […] Like the TANGO1/TANGO1-short PRDs, the cTAGE5 PRD also interacts with Sec23 (Goldberg, 2016; Saito et al., 2011; Wang et al., 2016).

Moreover, the following text is now included in the Discussion. “We have not tested whether cTAGE5, or for that matter TANGO1 short, can assemble into a ring in cells lacking TANGO1. We have not been able to create a form of cTAGE5 and TANGO1 short with a label or an antibody to visualize the domains proximal to the membranes, which makes it difficult to discern their location precisely even in the presence of endogenous TANGO1.”

5) Linear filaments in a planar membrane could generate rings, but association of the rings with one another must require multivalent interactions that go beyond head-to-tail filament formation. This point should be mentioned.

Now included in the text in the first description of a filament (first section of the Results, last paragraph, penultimate line). “We cannot rule out the possibility of multivalent interactions that go beyond head-to-tail filament formation.”

6) The authors propose that "TANGO1/cTAGE5 interactions with COPII […] act as an inward attractant, holding the filament in a ring-like configuration." But then why would the δ-PRD mutant still form rings?

It is important to note that these cells still contain TANGO1-short (Figure 1) and cTAGE5, both of which will recruit TANGO1ΔPRD to ERES.

Perhaps they mean to state that the COPII interaction constrains the size and placement of the rings.

We suggest that COPII interactions affects and not necessarily constrains/shrink the size.

7) The minimal TEER is the same 40 amino acids that are required for TANGO1 self-association. What does this mean? A crucial point that should be addressed is whether these two interactions are mutually exclusive.

The following statement is included in the text. “This tells us that the minimal TEER is exactly the same forty amino acids we identified in the previous section, as those required for the self-association of TANGO1. This implies that either a TANGO1 dimer can recruit a tether or the tether links two TANGO1 monomers. This hypothesis is tested and presented in Figure 7.”

8) Subsection “Compartment tethering in a TANGO1 ring assembly pathway”, tenth paragraph: the text refers to quantitation in Figure 6, but should it be Figure 7?

Yes, the reviewer is correct and we have changed this accordingly.

9) "In all cases, exit sites (as marked by TANGO1 and SEC31) are still recruited to intracellular collagen accumulations (Figure 7—figure supplement 1)." I don't see that result in Figure 7—figure supplement 1, and actually, I'm not sure what that supplemental figure is intended to show.

This is meant to show that exit sites (ERES) still form in cells depleted of tethers. We were requested to include this data in the previous review. We’ve now changed this description to state the following, “In all cases, exit sites, as marked by TANGO1 and SEC31, are still formed.”

10) Abstract: "for"? its self-association?

We have changed this sentence in the Abstract to the following. “Our data reveal that TANGO1 forms a ring organized by radial interaction with COPII, and lateral interactions with either cTAGE5 or itself.”

Subsection “Binding of TANGO1 to COPII controls TANGO1 ring formation”, second paragraph: TANGO1-short is not defined anywhere.

It is now in Figure 1 and described in the Introduction.

"CTAGE is on the periphery of rings." How many rings were counted to give confidence to this conclusion? How was this quantified?

See paragraph of the Discussion to clarify our data. “70 rings were manually counted from 12 cells and scored for cTAGE5 signal localisation within the ring. 21 rings showed peripherally located cTAGE5 while 49 had cTAGE5 within the ring formed by TANGO1.” This has been included in the figure legend for Figure 3.

"linear attraction along the length of the filament". By filament do they mean along the length of TANGO1? The authors should add a sentence stating that the interactions are consistent with. They have not defined the actual structural relationships here.

First, there is no mention of filament except in the Discussion. The statement referred by the reviewers is now changed as follows. “In our coarse-grained view of this fence of TANGO1 and TANGO1 family of proteins (cTAGE5 and TANGO1-short), we would describe our data thus far in terms of two general sets of interactions. First, lateral interactions mediated by TANGO1 self-association and its interaction with cTAGE5 and TANGO1-short, and second, inward attractions of TANGO1/cTAGE5/TANGO1-short to COPII, thus affecting the ring size and its placement with respect to COPII budding machinery.”

"has the potential to segregate cargo". This has not yet been shown.

We have changed this to the following. “Our new data describe a mechanism whereby the very processes by which TANGO1 recruits ERES machinery and cargo, also bring about its own assembly into a fence of defined size, which in turn remodels the ERES and in the lumen, via HSP47, binds and potentially segregates assembled cargo (Figure 8).”

The figures are changed as per the suggestions.